# Proteomic analysis shows decreased type I fibers and ectopic fat accumulation in skeletal muscle from women with PCOS

Elisabet Stener-Victorin[1], Gustaw Eriksson[1], Man Mohan Shrestha[2], Valentina Rodriguez Paris[3], Haojiang Lu[1], Jasmine Banks[3,4], Manisha Samad[2], Charlène Perian[2], Baptiste Jude[1], Viktor Engman[1], Roberto Boi[2], Emma Nilsson[5], Charlotte Ling[5], Jenny Nyström[2], Ingrid Wernstedt Asterholm[2], Nigel Turner[3,4], Johanna Lanner[1], Anna Benrick[2,6]*

[1]Department of Physiology and Pharmacology, Karolinska Institute, Stockholm, Sweden; [2]Department of Physiology, Institute of Neuroscience and Physiology, Sahlgrenska Academy, University of Gothenburg, Gothenburg, Sweden; [3]School of Biomedical Sciences, University of New South Wales, Sydney, Australia; [4]Victor Chang Cardiac Research Institute, Darlinghurst, Sydney, Australia; [5]Epigenetics and Diabetes Unit, Department of Clinical Sciences, Lund University Diabetes Centre, Lund University, Malmö, Sweden; [6]School of Health Sciences, University of Skövde, Skövde, Sweden

*For correspondence:
anna.benrick@gu.se

## Abstract

**Background:** Polycystic ovary syndrome's (PCOS) main feature is hyperandrogenism, which is linked to a higher risk of metabolic disorders. Gene expression analyses in adipose tissue and skeletal muscle reveal dysregulated metabolic pathways in women with PCOS, but these differences do not necessarily lead to changes in protein levels and biological function.

**Methods:** To advance our understanding of the molecular alterations in PCOS, we performed global proteomic and phosphorylation site analysis using tandem mass spectrometry, and analyzed gene expression and methylation. Adipose tissue and skeletal muscle were collected at baseline from 10 women with and without PCOS, and in women with PCOS after 5 weeks of treatment with electrical stimulation.

**Results:** Perilipin-1, a protein that typically coats the surface of lipid droplets in adipocytes, was increased whereas proteins involved in muscle contraction and type I muscle fiber function were downregulated in PCOS muscle. Proteins in the thick and thin filaments had many altered phosphorylation sites, indicating differences in protein activity and function. A mouse model was used to corroborate that androgen exposure leads to a shift in muscle fiber type in controls but not in skeletal muscle-specific androgen receptor knockout mice. The upregulated proteins in muscle post treatment were enriched in pathways involved in extracellular matrix organization and wound healing, which may reflect a protective adaptation to repeated contractions and tissue damage due to needling. A similar, albeit less pronounced, upregulation in extracellular matrix organization pathways was also seen in adipose tissue.

**Conclusions:** Our results suggest that hyperandrogenic women with PCOS have higher levels of extra-myocellular lipids and fewer oxidative insulin-sensitive type I muscle fibers. These could be key factors leading to insulin resistance in PCOS muscle while electric stimulation-induced tissue remodeling may be protective.

**Funding:** Swedish Research Council (2020-02485, 2022-00550, 2020-01463), Novo Nordisk Foundation (NNF22OC0072904), and IngaBritt and Arne Lundberg Foundation. Clinical trial number NTC01457209.

### eLife assessment

This **important** work employed global proteomic and phosphorylation site analysis to examine adipose tissue and skeletal muscle samples collected at baseline from a sample of 10 women, including those with and without PCOS, both before and after 5 weeks of electrical stimulation treatment. This work significantly enhances our knowledge by demonstrating that women with PCOS who exhibit protein hyperandrogenicity have elevated extramyocellular lipid levels and a decreased number of oxidative insulin-sensitive type I muscle fibers. The **convincing** evidence supporting these conclusions makes this research of broad interest not only to scientists but also to clinicians.

## Introduction

Polycystic ovary syndrome (PCOS) is a metabolic and endocrine disorder characterized by clinical signs of hyperandrogenism and reproductive dysfunction (*Joham et al., 2022*). Although not part of the diagnosis, insulin resistance and abdominal obesity are common and lead to an increased risk of type 2 diabetes (*Kakoly et al., 2019*). PCOS affects 8–17% of women worldwide. More than 50% of women with PCOS are overweight or obese, and obesity worsens all symptoms, including insulin resistance (*Kakoly et al., 2019*; *Barber and Franks, 2021*). In those with PCOS, obesity is associated with an altered adipose tissue function, increased visceral fat, adipocyte hypertrophy, and lower adiponectin levels compared with BMI-matched controls (*Mannerås-Holm et al., 2011*; *Villa and Pratley, 2011*; *Mannerås-Holm et al., 2014*). In recent independent bidirectional Mendelian studies, obesity is even thought to contribute to or cause the development of PCOS (*Brower et al., 2019*; *Zhao et al., 2020*; *Liu et al., 2022*). Adipose tissue dysfunction also leads to consequences in other metabolic tissues. Overweight/obese women with PCOS are at increased risk of developing lipotoxicity due to increased triglyceride and free fatty acid levels and increased fatty acid uptake into non-adipose cells, including skeletal muscle, which is exacerbated by increased intra-abdominal fat with high lipolytic activity (*Dumesic et al., 2022*). The lipotoxic state in skeletal muscle includes increased expression of genes involved in lipid storage and epigenetic changes, as demonstrated in a PCOS-like sheep model (*Guo et al., 2020*). Consequently, excessive uptake and extracellular storage of free fatty acid in skeletal muscle promotes insulin resistance. Moreover, skeletal muscle from women with PCOS exhibits transcriptional, epigenetic, and protein changes associated with insulin resistance, accompanied by an inflammatory, oxidative, and lipotoxic state (*Corbould et al., 2005*; *Skov et al., 2007*; *Nilsson et al., 2018*; *Stepto et al., 2019*; *Manti et al., 2020*; *Stepto et al., 2020*; *Moreno-Asso et al., 2021*).

Several studies show that defects in mRNA expression of mitochondrial function, fat oxidation, and immunometabolic pathways in adipose tissue and skeletal muscle contribute to insulin resistance in women with PCOS (*Skov et al., 2007*; *Nilsson et al., 2018*; *Stepto et al., 2019*; *Manti et al., 2020*; *Stepto et al., 2020*). Interestingly, the mechanism underlying the reduced insulin-stimulated glucose uptake in PCOS skeletal muscle likely differs from insulin resistance in BMI-matched controls (*Corbould et al., 2005*; *Nilsson et al., 2018*). This finding is consistent with the theory that PCOS is a disorder with genetically distinct subtypes of PCOS (*Dapas et al., 2020*). However, mRNA expression does not always translate to alterations in protein levels and changes in biological function. The first proteomics data on visceral fat and skeletal muscle from severely obese women with PCOS used mass spectrometry of selected protein spots on a 2D gel electrophoresis and were published a decade ago (*Cortón et al., 2008*; *Montes-Nieto et al., 2013*; *Insenser et al., 2016*). This method is limited as only a few proteins could be identified. Today, the combination of liquid chromatography with tandem mass spectrometry provides us with thousands of proteins. Recent publications map the proteome in serum, follicular fluid, ovary, and endometrium from women with PCOS, and the proteome in serum may provide new biomarkers for PCOS (*Li et al., 2020*; *Abdulkhalikova et al., 2022*; *Wang et al., 2022*). We here use a nontargeted quantitative proteomics approach to advance our understanding of the pathophysiology in adipose tissue and skeletal muscle from women with PCOS.

We have previously shown that electrically stimulated muscle contractions, known as electroacupuncture, improve glucose regulation and decrease androgen levels in overweight/obese women with PCOS (*Stener-Victorin et al., 2016*). When acupuncture needles are stimulated by low-frequency electrical stimulation, they cause muscle contractions similar to those that occur during exercise. Muscle contractions activate specific physiological signaling pathways, and electrical muscle stimulations and exercise act through partially similar signaling pathways to induce glucose uptake in the acute response to muscle contractions (*Benrick et al., 2020*). In addition, long-term intervention with both exercise and electrical stimulation has been shown to improve glucose regulation, promote ovulation, decrease muscle sympathetic nerve activity, and reduce circulating androgens in women with PCOS (*Stener-Victorin et al., 2009*; *Hutchison et al., 2011*; *Jedel et al., 2011*; *Johansson et al., 2013*; *Stener-Victorin et al., 2016*; *Tiwari et al., 2019*). The molecular mechanisms mediating the effect on adipose tissue and skeletal muscle in response to long-term electrical stimulation remain unclear. Therefore, the second aim is to use proteomics to provide mechanistic explanations for the improved glucose homeostasis in response to long-term electrical stimulation treatment.

## Methods
### Ethical approval
The study was conducted at the Sahlgrenska University Hospital and the Sahlgrenska Academy, University of Gothenburg, Gothenburg, Sweden, in accordance with the standards set by the *Declaration of Helsinki*. Procedures have been approved by the Regional Ethical Review Board of the University of Gothenburg (approval number 520-11) and the study was registered at https://clinicaltrials.gov/ (NTC01457209). All women provided oral and written informed consent before participation in the study.

### Participants and study protocol
Women with PCOS were diagnosed according to the Rotterdam criteria (*Group. REA-SPCW, 2004*). This cohort has previously been described in detail and included 21 overweight and obese cases and 21 overweight and obese controls matched for age, weight, and BMI (*Kokosar et al., 2016*; *Stener-Victorin et al., 2016*). Of these, subcutaneous adipose tissue and skeletal muscle biopsies from 10 women with PCOS and 10 controls were included in the proteomics analysis. Participants reported to the laboratory in the morning after an overnight fast on menstrual cycle days 1–10 or irrespective of cycle day in ameno/oligomenorrheic women. Anthropometrics were measured and BMI was calculated as previously described (*Kokosar et al., 2016*; *Stener-Victorin et al., 2016*). Fasting

**Table 1.** Anthropometric and biochemical analyses in study participants.

| Variable | Controls baseline (n=10) | PCOS baseline (n=10) | Controls vs. PCOS (p-value) | PCOS treatment (n=10) | Baseline vs. treatment (p-value) |
|---|---|---|---|---|---|
| Age (years) | 31.2±4.8 | 30.1±5.1 | 0.60 | | |
| Weight (kg) | 88.2±11.0 | 88.6±15.6 | 0.82 | 89.0±14.9 | 0.33 |
| BMI (kg/m$^2$) | 30.5±3.7 | 30.5±4.1 | 0.88 | 30.7±4.0 | 0.32 |
| Ferriman-Gallwey score | 2.1±1.7 | 7.7±4.5 | **0.003** | 9.2±6.0 | 0.48 |
| Testosterone (pg/ml) | 258±77 | 476±211 | **0.004** | 347±220 | **0.012** |
| Insulin (mU/ml) | 8.2±3.5 | 11.4±8.5 | 0.47 | 8.5±4.8 | 0.11 |
| Glucose (mmol/l) | 5.2±0.3 | 4.8±0.4 | **0.044** | 4.8±0.3 | 0.67 |
| HOMA-IR | 1.92±0.96 | 2.75±2.00 | 0.41 | 1.89±1.33 | **0.037** |
| Hemoglobin A1c (mmol/mol) | 31.5±2.6 | 31.7±2.7 | 1.00 | 30.5±2.9 | **0.035** |
| Triglycerides (mmol/l) | 0.71±0.18 | 1.11±0.58 | 0.076 | 1.01±0.58 | 0.066 |

Data are presented as mean ± SD. Differences between PCOS and controls were analyzed by Mann-Whitney *U* test. Wilcoxon signed-rank test was used to analyze changes between measurements at baseline and after 5 weeks of treatment. p-value < 0.05 was considered significant.
*

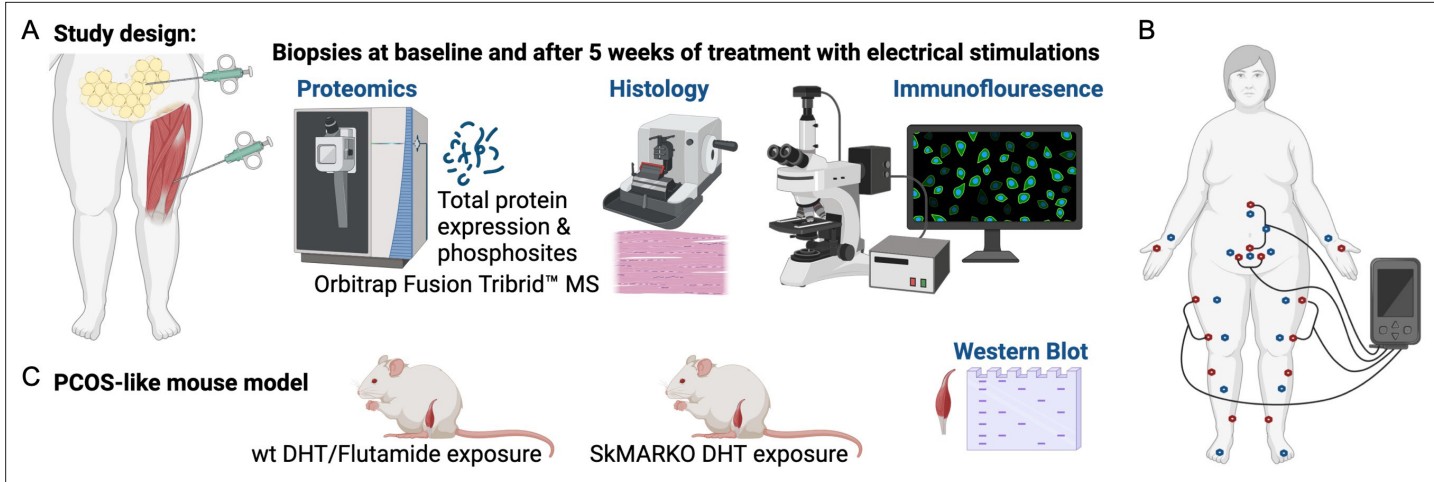

**Figure 1.** Study design. (**A**) Muscle and fat biopsies collected from 10 controls and 10 women with polycystic ovary syndrome (PCOS) at baseline and after treatment with electrical stimulations. Electrical stimulations were given 3 times/week for 5 weeks. (**B**) The electrical stimulation protocol alternating between protocol 1 in red dots and protocol 2 in blue dots. Acupuncture points not connected to the stimulator were stimulated manually. (**C**) A PCOS-like mouse model treated with the androgen receptor blocker flutamide or lacking androgen receptors in skeletal muscle (SkMARKO). Created with https://www.biorender.com/.

blood samples were taken and HOMA-IR (fasting insulin [mU/ml] × fasting glucose [mM]/22.5) was calculated. Circulating testosterone was measured by GC-MS/MS (*Kokosar et al., 2016*). Anthropometrics, reproductive and endocrine variables for those included in the proteomics analysis are given in *Table 1*. Needle biopsies from subcutaneous adipose tissue from the umbilical area and skeletal muscle tissue from vastus lateralis were obtained under local anesthesia (Xylocaine, Astra-Zeneca AB, Södertälje, Sweden), from cases and controls (*Figure 1A*). The fat biopsies were rinsed with saline before both tissues were snap-frozen in liquid nitrogen and stored at −80°C until further analysis. Thereafter, women with PCOS received low-frequency electrical stimulations causing muscle contractions, so called electroacupuncture. Acupuncture needles were placed in somatic segments corresponding to the innervation of the ovaries and pancreas, bilaterally in abdominal muscle, in quadriceps muscles, and in the muscles below the medial side of the knee. Needles were inserted to a depth of 15–40 mm with the aim of reaching the muscles. Needles were connected to an electrical stimulator (CEFAR ACUS 4; Cefar-Compex Scandinavia, Landsbro, Sweden) and stimulated with a low-frequency (2 Hz) electrical signal for 30 min. The intensity was adjusted every 10th minute due to receptor adaptation, with the intention to produce local muscle contractions without pain or discomfort. Treatment was given three times per week over 5 weeks, and the number of treatments varied from 11 to 19. Two sets of needle placements were alternated to avoid soreness (*Figure 1B*; *Stener-Victorin et al., 2016*). Baseline measurements were repeated after 5 weeks of treatment, within 48 hr after the last treatment, and new fat and muscle biopsies were collected. After all relevant clinical information was obtained, samples were coded and anonymized.

## Proteomic sample preparation and LC-MS/MS analysis
### Tissue lysis and LC/MS-MS
The individual muscle and adipose tissue biopsies (20–50 mg) were homogenized in 300 µl lysis buffer (50 mM triethylammonium bicarbonate [TEAB; Fluka, Sigma-Aldrich] and 2% sodium dodecyl sulfate) with 1.4 mm ceramic spheres (FastPrep matrix D) using FastPrep-24 instrument (MP Biomedicals, OH, USA). The protein concentration was determined using Pierce BCA Protein Assay (Thermo Scientific, Rockford, IL, USA) and the Benchmark Plus microplate reader (Bio-Rad Laboratories, Hercules, CA, USA).

### Sample preparation for global proteomic analysis
Aliquots containing 25 µg of each individual sample were digested with trypsin using the filter-aided sample preparation method. Briefly, protein samples were reduced with 100 mM dithiothreitol at 60°C

for 30 min, transferred on 30 kDa MWCO Nanosep centrifugal filters (Pall Life Sciences, Ann Arbor, MI, USA), washed with 8 M urea solution and alkylated with 10 mM methyl methanethiosulfonate in 50 mM TEAB and 1% sodium deoxycholate. Digestion was performed in 50 mM TEAB, 1% sodium deoxycholate at 37°C in two stages: the samples were incubated with 250 ng of Pierce MS-grade trypsin (Thermo Fisher Scientific, Rockford, IL, USA) for 3 hr, then 250 ng more of trypsin was added and the digestion was performed overnight. The peptides were collected by centrifugation, labeled using TMT 10-plex isobaric mass tagging reagents (Thermo Scientific). Sodium deoxycholate was then removed by acidification with 10% trifluoroacetic acid. The mixed labeled samples were fractionated on the AKTA chromatography system (GE Healthcare Life Sciences, Sweden) using the XBridge C18 3.5 μm, 3.0×150 mm column (Waters Corporation, Milford, CT, USA) and 25 min gradient from 7% to 40% solvent B at the flow rate of 0.4 ml/min; solvent A was 10 mM ammonium formate in water at pH 10.00, solvent B was 90% acetonitrile, 10% 10 mM ammonium formate in water at pH 10.00. The initial 40 fractions were combined into 20 pooled fractions in the order 1+21, 2+22, 3+23, etc. The pooled fractions were dried on Speedvac and reconstituted in 20 μl of 3% acetonitrile, 0.1% formic acid for analysis.

## Sample preparation for phosphoproteomic analysis

Aliquots containing 450 μg of each individual sample were digested with trypsin using the filter-aided sample preparation method. The phosphopeptides were enriched using Pierce TiO$_2$ Phosphopeptide Enrichment and Clean Up Kit (Thermo Fisher Scientific). The purified phosphopeptide samples were evaporated to dryness, reconstituted in 50 mM TEAB, and labeled using TMT 10-plex isobaric mass tagging reagents (Thermo Fisher Scientific). The TMT-labeled phosphopeptide samples were mixed into corresponding sets and purified using Pierce C-18 Spin Columns (Thermo Fisher Scientific). Purified samples were dried on Speedvac and reconstituted in 15 μl of 3% acetonitrile, 0.1% formic acid for analysis.

## LC-MS/MS analysis

All samples were analyzed on Orbitrap Fusion Tribrid (Thermo Fisher Scientific) interfaced with Thermo Easy-nLC 1000 nanoflow liquid chromatography system (Thermo Fisher Scientific). Peptides were trapped on the C18 trap column (100 μm × 3 cm, particle size 3 μm) separated on the C18 analytical column (75 μm × 30 cm) home-packed with 3 μm Reprosil-Pur C18-AQ particles (Dr. Maisch, Germany) using the gradient from 5% to 25% B in 45 min, from 25% to 80% B in 5 min, solvent A was 0.2% formic acid and solvent B was 98% acetonitrile, 0.2% formic acid. Precursor ion mass spectra were recorded at 120,000 resolution. The most intense precursor ions were selected ('top speed' setting with a duty cycle of 3 s), fragmented using CID at collision energy setting of 30 spectra and the MS2 spectra were recorded in ion trap. Dynamic exclusion was set to 30 s with 10 ppm tolerance. MS3 spectra were recorded at 60,000 resolution with HCD fragmentation at a collision energy of 55 using the synchronous precursor selection of the five most abundant MS/MS fragments. The phosphopeptides were trapped on the NanoViper C18 trap column (100 μm × 2 cm, particle size 2 μm, Thermo Scientific) and separated on the home-packed C18 analytical column (75 μm × 30 cm) using the gradient from 7% to 32% B in 100 min, from 32% to 100% B in 5 min, solvent A was 0.2% formic acid and solvent B was 80% acetonitrile, 0.2% formic acid. The mass spectrometry settings were the same as described above for the global proteomic analysis, but HCD fragmentation at collision energy of 33 was used in MS2.

## Proteomic data analysis

Identification was performed using Proteome Discoverer version 2.4 (Thermo Fisher Scientific). The database search was performed using the Mascot search engine v. 2.5.1 (Matrix Science, London, UK) against the Swiss-Prot *Homo sapiens* database. For phosphopeptide samples, phosphorylation on serine, threonine, and tyrosine was added as a variable modification. Quantification was performed in Proteome Discoverer 2.4. TMT reporter ions were identified with 3 mmu mass tolerance in the MS3 HCD spectra for the total proteome experiment and with 20 ppm mass tolerance in the MS2 HCD spectra for the phosphopeptide experiment, and the TMT reporter S/N values for each sample were normalized within Proteome Discoverer 2.4 on the total peptide amount. Only the unique identified peptides were considered for protein quantification. The mass spectrometry proteomics data

have been deposited to the ProteomeXchange Consortium via the Proteomics Identifications (PRIDE) (RRID:SCR_003411) (*Deutsch et al., 2020*) partner repository with the dataset identifier PXD025358.

The normalized abundance counts were used for the downstream analysis using the Differential Enrichment analysis of Proteomics data package (DEP) (*Zhang et al., 2018*). The counts were $\log_2$ transformed and proteins that were quantified in less than 2/3 of the samples were removed. Missing values were imputed using random draws from a Gaussian distribution around the minimal value as the missing value was not random but concentrated to proteins with low intensities. The data was batch effect adjusted using ComBat (*Johnson et al., 2007*) with the LC-MS/MS run assigned as the batch covariate. Differential expression analysis was performed on the dataset with DEP's test_diff function which uses protein-wise linear models and empirical Bayes statistics using limma (*Smyth, 2004*). The differential expression analysis calculated the $\log_2$ fold changes (FC), *p-values* and *q-values* between the three groups control and PCOS at baseline (W0), and PCOS after 5 weeks of treatment (W5) generating two different result datasets. Proteins and phosphorylation enrichment were determined to be significantly differentially expressed between the groups if the p-value <0.05, and $\log_2$ FC $\geq 0.5$ or $\log_2$ FC $\leq -0.5$.

## mRNA expression and DNA methylation arrays

mRNA was extracted from adipose tissue (n=17) and skeletal muscle (n=8) biopsies collected at steady state during the hyperinsulinemic-euglycemic clamp before and after 5 weeks of electrical stimulation in those with PCOS using the RNeasy Lipid Tissue Mini Kit for adipose tissue and RNeasy Fibrous Tissue Mini Kit for skeletal muscle (QIAGEN). Nucleic acid concentration was measured with a spectrophotometer (NanoDrop, Thermo Scientific), and RNA quality was determined with an automated electrophoresis station (Experion, Bio-Rad). A HumanHT-12 v4 Expression BeadChip array (Illumina) was used to analyze global mRNA expression. cRNA synthesis, including biotin labeling, was carried out using an Illumina TotalPrep RNA Amplification Kit (Life Technologies and Invitrogen). Biotin-cRNA complex was then fragmented and hybridized to the probes on the Illumina BeadChip array before being hybridized and stained with streptavidin-Cy3 according to the manufacturer's instructions. Probes were visualized with an Illumina HiScan fluorescence camera. The Oligo package from Bioconductor was used to compute robust multichip average expression measures (*Bolstad et al., 2003*).

For methylation array studies, DNA was isolated from adipose tissue (n=17) and skeletal muscle (n=9) biopsies taken at steady state during the hyperinsulinemic-euglycemic clamp before and after 5 weeks of electrical stimulation of women with PCOS using the QIAamp DNA Mini Kit (QIAGEN). Nucleic acid concentrations and purity were estimated with a NanoDrop spectrophotometer (Thermo Scientific, Wilmington, DE, USA), and DNA integrity was checked by gel electrophoresis. Genome-wide DNA methylation was analyzed with the Infinium HumanMethylation450k BeadChip array (Illumina). The array contains 485,577 cytosine probes covering 21,231 (99%) RefSeq genes (*Bibikova et al., 2011*). A DNA Methylation Kit (D5001-D5002, Zymo Research) was used to convert genomic DNA to bisulfite-modified DNA. Briefly, gDNA (500 ng) of high quality was fragmented and hybridized on the BeadChip, and the intensities of the signals were measured with a HiScanQ scanner (Illumina).

### Array data analysis

The bioinformatics analyses of DNA methylation array data were performed as described previously (*Rönn et al., 2015*). In brief, Y chromosome probes, rs-probes, and probes with an average detection p-value>0.01 were removed. After quality control and filtering, methylation data were obtained for 298,289 CpG sites in adipose tissue and 298,332 CpG sites in skeletal muscle. Beta-values were converted to M-values, $M=\log_2(\beta/(1-\beta))$, which were used for all data analyses. Data were then quantile-normalized and batch-corrected with COMBAT (*Johnson et al., 2007*). The differentially methylated sites were identified using a paired t-test (limma package, Bioconductor). To improve interpretation, after all the preprocessing steps, the data were reconverted to beta-values ranging from 0% (unmethylated) to 100% (completely methylated).

## Pathway enrichment analysis

The tool Uniprot, Universal Protein Resource (RRID:SCR_002380), was used to retrieve protein names. We applied enrichment analysis to all differently expressed proteins and phosphorylation sites using

Enrichr (RRID:SCR_001575) and STRING (RRID:SCR_005223). Ontology terms with a q-value <0.05 and including at least 3 proteins/phosphosites or an odds ratio >100 were considered as enriched.

## Histological analyses and immunofluorescence

Skeletal muscle and adipose tissue biopsies were fixed in histofix (Histolab, Sweden) for >72 hr and then stored in 70% ethanol. Tissues were dehydrated and fixated in paraffin blocks. Paraffin-embedded adipose tissue and muscle tissue were cut into 7 µm sections using a rotary microtome (Leica Micro-tome) and mounted on Superfrost Plus Adhesion microscope glass slides (Epredia J1800AMNZ, #10149870, Thermo Fisher Scientific). Picrosirius red staining (cat#24901-250, Polysciences, Inc) was used to identify and quantify fibrillar collagen in adipose and muscle tissue. Adipose tissue quantification of picrosirius red staining before and after electrical stimulation treatment was performed using a semi-automatic macro in ImageJ software. This macro allows for calculation of the total area ($\mu m^2$) and the % of collagen staining from each area adjusting the minimum and maximum thresholds. Three different random pictures per section (4–5 sections/subject) were taken at ×10 or ×20 magnification using a regular bright-field microscope (Olympus BX60 and PlanApo, ×20/0.7, Olympus, Japan). All images were analyzed on ImageJ software v1.47 (National Institutes of Health, Bethesda, MD, USA) using this protocol with the following modification: threshold min 0, max 2. Skeletal muscle quantification of picrosirius red staining was performed using the same protocol described above. % of collagen staining was calculated on 8–10 images of different microscopic fields from each muscle sample.

For immunofluorescence, the muscle sections were deparaffinized twice in xylene (#534056, Sigma-Aldrich) for 5 min. Sections were rehydrated stepwise twice in 100% ethanol and once in 95%, 70%, 50% ethanol and in deionized water for 5 min each before a final rinse in PBS (#18912-014, Gibco, pH 7.4) for 5 min. The slides were subjected to heat-induced antigen retrieval by heating in antigen retrieval buffer (10 mM citric acid monohydrate, 0.05% [vol/vol] Tween-20, pH 6.0) until it reaches the boiling point and cooling to room temperature. The tissue sections were incubated in blocking buffer (3% normal donkey serum [NDS] [vol/vol] in PBS) at room temperature for 1 hr followed by overnight incubation at 4°C with primary antibodies (rabbit anti-perilipin-1, dilution 1:150 [Abcam Cat# ab3526] and mouse anti-myosin [skeletal slow], dilution 1:300 [MYH7 antibody, Sigma-Aldrich Cat# M8421]) diluted in incubation buffer (PBS containing 0.3% Triton X-100, 1% BSA, 1% NDS, and 0.01% sodium azide, pH 7.2). After rinsing the slides three times for 10 min each in PBS, the sections were incubated with fluorochrome-conjugated secondary antibodies (Donkey anti-Rabbit IgG [H+L] Highly Cross-Absorbed Secondary Antibody, Alexa Fluor 555, diluted 1:250 [Thermo Fisher Scientific Cat# A-31572] and Donkey anti-Mouse IgG [H+L] Highly Cross-Absorbed Secondary Antibody, Alexa Fluor Plus 448, diluted 1:250 [Thermo Fisher Scientific Cat# A32766]) diluted in incubation buffer. The muscle sections were rinsed three times for 10 min each in PBS. The slides were mounted with coverslips (#ECN 631-1574, VWR) using Vectashield antifade mounting medium with DAPI (#H-1200, Vectashield). Images were obtained using a Zeiss LSM 700, AxioObserver microscope with Plan-Apochromat ×10/0.45 M27 objective lens. Argon lasers of 488 nm and 555 nm wavelengths were used to excite Alexa Fluor 488 (green) and Alexa Fluor 555 (red) respectively, and Laser Diode 405 nm to excite DAPI (blue). Quantification of perilipin-1 expression in skeletal muscle cells from control and PCOS groups was performed using ImageJ software (National Institutes of Health, Bethesda, MD, USA). The channels of the images were split and converted into 8-bit. The minimum and maximum thresholds were adjusted and kept constant for all the images. Regions of interest were drawn around the cells and empty space for background intensity measurement. The mean perilipin-1 intensity was measured and corrected by deducting the background. A total of 28 PCOS and 33 control cells were quantified.

Skeletal muscle fiber size and type were quantified in muscle biopsies frozen in Tissue-Tek O.C.T. Compound (Sakura Finetek, Gothenburg, Sweden). Cross sections (10 µm) were cryosectioned using an NX70-Epredia cryostat, moved onto glass slides (Expredia, J1800AMNZ) and stored at –20°C. The sections were subsequently immunohistochemically stained for type I fibers and fiber boundaries. In brief, the sections were dried at room temperature for 60 min and then fixed in 4% formalde-hyde (Merck, 100496) for 30 min, permeabilized with 0.5% PBS-Triton X-100 (Sigma-Aldrich, 9036-19-5) for 20 min and thereafter incubated with 0.25% PBS-Triton X-100 with 10% goat serum for 30 min. The sections were then incubated with primary MYH7 antibody (1:25; DSHB, BA-F8) for type I fibers overnight at 4°C and subsequently secondary antibody Alexa Flour 568 (1:500; Thermo Fisher,

A-11031) for 60 min at room temperature, both in 0.1% PBS-Triton X-100 with 1% BSA. Finally, the sections were incubated with WGA Oregon Green 488 (Invitrogen, W7024) for fiber boundaries for 3 hr, whereafter Fluoromount-G mounting medium (Thermo Fisher, 00-4959-52) and coverslips were applied. The slides were visualized using a Zeiss AxioScan.Z1 slide scanner. Fiber cross-sectional area was automatically determined using MyoVision v1.0 and the proportion of type I fibers was manually counted on ImageJ. A total of 579 fibers from seven controls (60–150 fibers per muscle section) and 177 fibers (15–80 fibers per muscle section) from women with PCOS were quantified. Data are graphically depicted with each individual fiber quantified.

## Mouse study protocol and western blot analysis

All animal experiments were carried out in compliance with the ARRIVE guidelines. Three-week-old wild-type (wt) female mice on C57Bl/6J background were purchased from Janvier Labs (C57BL/6NRj, Le Genest-Saint-Isle, France). Female skeletal muscle androgen receptor knockout mice (SkMARKO) were generated by crossing ARflox mice with B6;C3-Tg(ACTA1-rtTA,tetO-cre)102Monk/J (HSA-rtTA/TRE-Cre) mice (*Xiong et al., 2022*). To induce Cre recombinase expression, at 3 weeks of age, SkMARKO mice were given a diet containing 200 mg/kg doxycycline (Specialty Feeds SF11-059) for the entire duration of the experiment. Mice were maintained under standard housing conditions; ad libitum access to food and water in a temperature- and humidity-controlled, 12 hr light/dark environment. Procedures were approved by the Sydney Local Health District Animal Welfare Committee within National Health and Medical Research Council guidelines for animal experimentation or the Stockholm Ethical Committee for Animal Research (approval number 20485-2020). At 4 weeks of age, wt and SkMARKO female mice received a subcutaneous silastic implant containing 5–10 mg DHT (5α-Androstan-17β-ol-3-one, A8380, Sigma-Aldrich, St. Louis, MO, USA) or empty implant (n=5–8/group). A subset of DHT-exposed wt mice received a slow-release flutamide pellet (n=5, 25 mg flutamide/pellet, 90-day release, Innovative Research of America, Sarasota, FL, USA) (*Figure 1C*; *Ascani et al., 2023*). At 15–17 weeks of age, the mice were euthanized and gastrocnemius muscle tissue was dissected and snap-frozen.

15–20 mg of gastrocnemius muscle was homogenized in RIPA buffer along with protease inhibitors. Protein was quantified using Pierce BCA assay (Thermo Fisher Scientific, Cat# 23227). Diluted protein lysates were mixed with loading buffer containing β-mercaptoethanol, and heated at 65°C, before being loaded into polyacrylamide gel (handcast 10% or AnykD PROTEAN TGX Precast Protein Stain-Free Gel [Bio-Rad, CA, USA]) and electro-transferred to PVDF membranes using a PVDF Transfer Pack. The membranes were blocked with blocking solution (5% BSA or 5% skim milk in TBS containing 0.1% Tween 20) for 1 hr and incubated overnight with anti-myosin primary antibody, dilution 1:1000 (MYH7 antibody, Sigma-Aldrich Cat# M8421). After washing and incubation with secondary antibody, dilution 1:10,000 (rabbit anti-mouse IgG HRP, Abcam Cat# 97046), immunoreactive protein bands were visualized through enhanced chemiluminescence using ECL substrate. Bands were visualized with the ChemiDoc XRS system (Bio-Rad, CA, USA) and analyzed with the image analysis program Image Lab (Bio-Rad, CA, USA). After initial imaging, membranes were stripped in mild stripping buffer, blocked, and re-probed with GAPDH (Abcam, Cat# ab8245) or beta-actin antibody (Santa Cruz, Cat# 47778).

## Statistics

Differences in clinical characteristics and histological quantification between women with PCOS and controls were assessed using the Mann-Whitney U test and data are presented as mean ± SD. Wilcoxon signed-rank test was used to analyze changes between measurements at baseline and after 5 weeks of treatment. Differences in protein expression were calculated on $\log_2$ fold change. Proteins and phosphorylation enrichments were determined to be significantly differentially expressed between cases and controls, and after 5 weeks of treatment in women with PCOS, if the p-value <0.05, and $\log_2$ FC ≥0.5 or $\log_2$ FC ≤–0.5. Proteomic data are presented as $\log_2$ fold change. Differences between wt controls and treated mice were assessed using one-way ANOVA with Dunnet's multiple comparisons test. A two-way ANOVA was used to analyze the effect of treatment and mouse genotype, and data are presented as mean ± SEM. No statistical methods were used to predetermine sample size; it was based on previous experience. Animals were allocated to experimental groups arbitrarily without formal randomization.

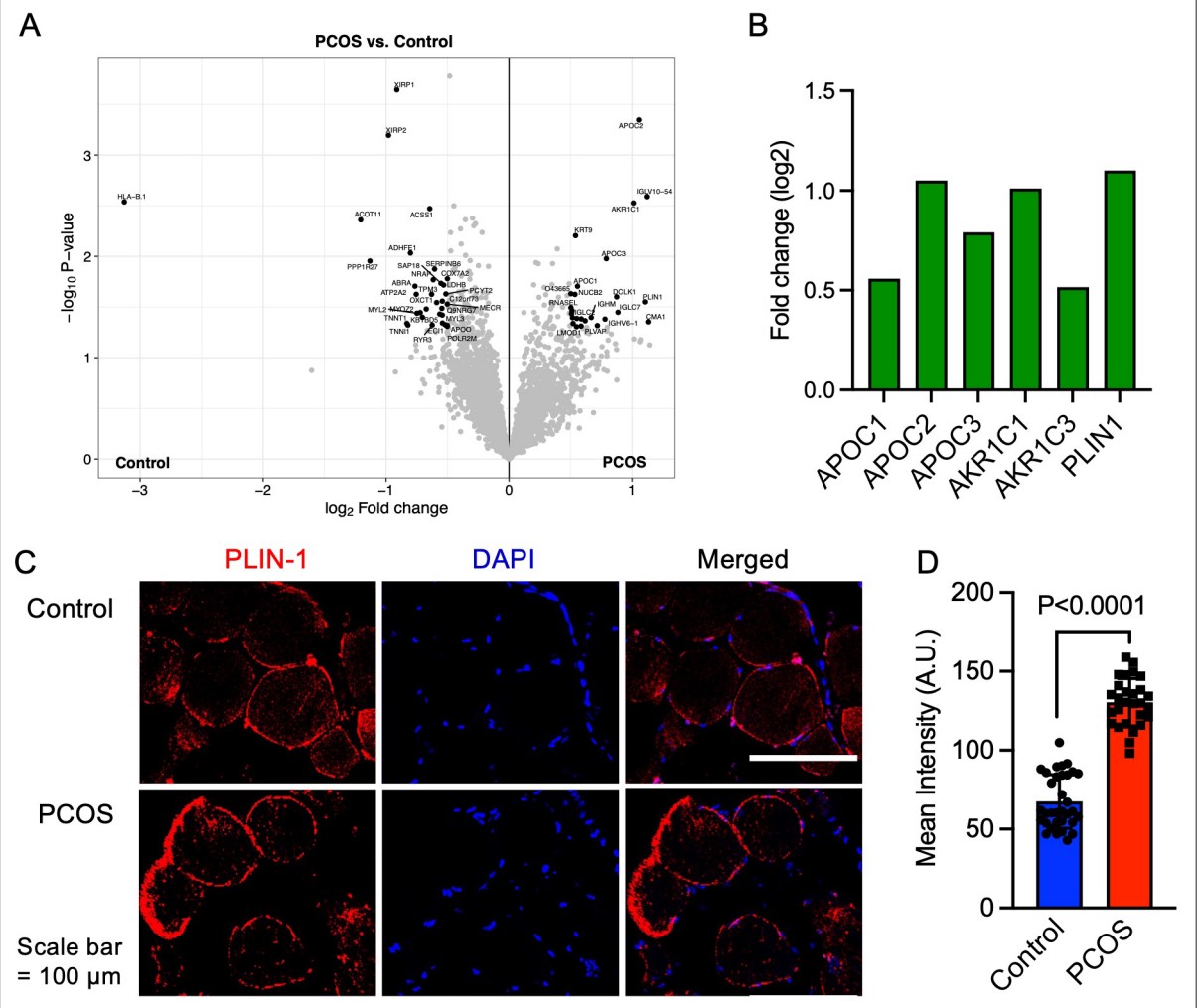

**Figure 2.** Protein expression and upregulated proteins in skeletal muscle. (**A**) Volcano plot showing the mean protein log$_2$ fold change in skeletal muscle (polycystic ovary syndrome [PCOS] vs controls) using limma method, and plotted against the −log$_{10}$ p-value highlighting significantly regulated proteins in black (p<0.05, log$_2$ fold change ± 0.5), n=10/group. (**B**) Increased protein expression of apolipoproteins C1 and C2, aldo-keto reductase (AKR) family 1 C1 and C3, and perilipin-1 in those with PCOS, (**C**) staining of perilipin-1 and DAPI in skeletal muscle, (**D**) quantification of perilipin-1 staining in skeletal muscle cells from control (n=33) and PCOS (n=28). Difference is based on Mann-Whitney U test and data are presented as mean ± SD.

## Results
### Clinical characteristics

Women with PCOS had more antral follicles <9 mm (22.7±7.9 vs 9.4±4.1, p=0.001), larger ovary volume (8.0±2.9 vs 5.0±2.7 ml, p=0.028), higher Ferriman-Gallwey score, and higher circulating testosterone than controls (*Table 1*). Six of the 10 women with PCOS met all three PCOS criteria; two had hyperandrogenemia and PCO morphology, one had hyperandrogenism and irregular cycles, and one had irregular cycles and PCO morphology. Five weeks of treatment lowered testosterone, HOMA-IR, and HbA1c levels, tended to decrease triglyceride levels but did not improve Ferriman-Gallwey score (*Table 1*).

Data are presented as mean ± SD. Differences between PCOS and controls were analyzed by Mann-Whitney U test. Wilcoxon signed-rank test was used to analyze changes between measurements at baseline and after 5 weeks of treatment.

### Total protein expression and phosphorylation in skeletal muscle

In total, we identified 3480 proteins in skeletal muscle. 58 unique proteins were found to be differentially expressed in skeletal muscle from women with PCOS versus controls (p<0.05, and log$_2$

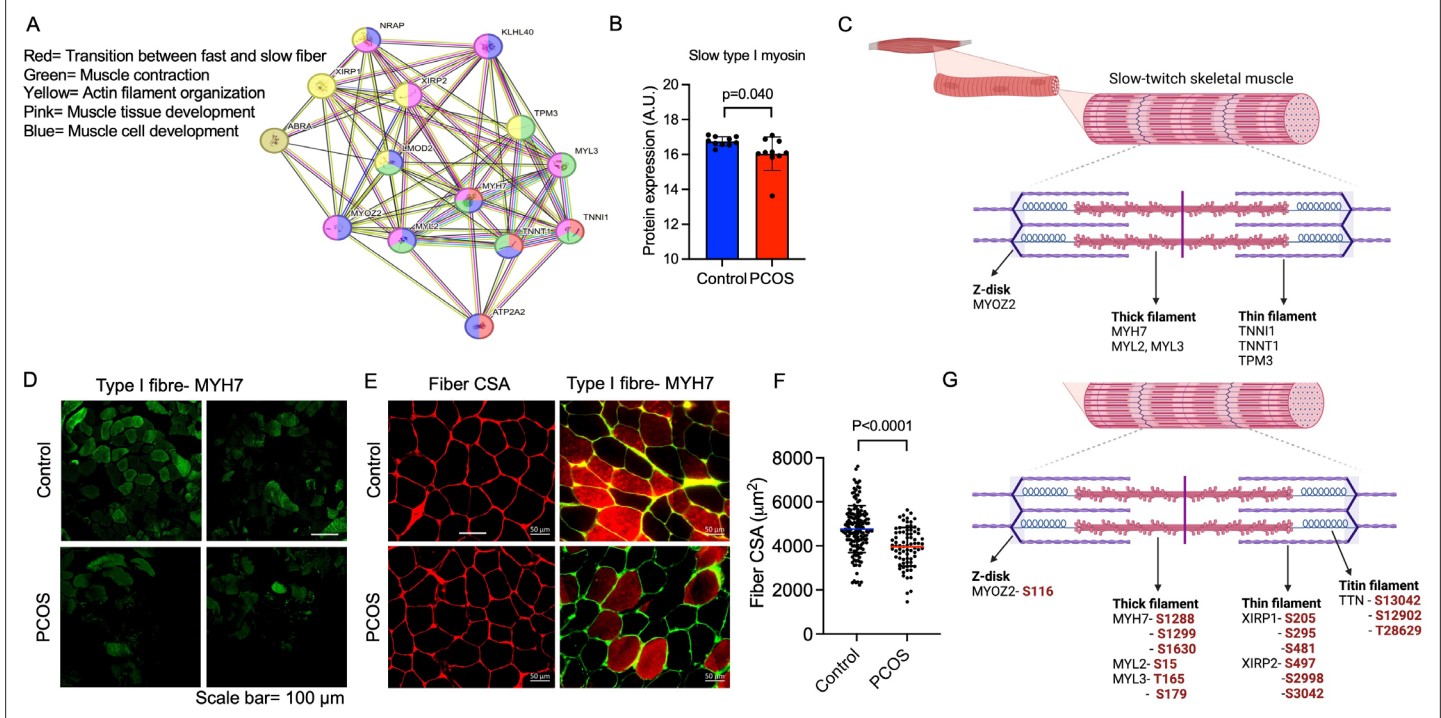

**Figure 3.** Enriched downregulated pathways involved in muscle contraction and transition between fast and slow fibers in PCOS. (**A**) Protein network on proteins with a lower expression in polycystic ovary syndrome (PCOS) skeletal muscle vs. controls. Lines indicate protein-protein associations. (**B**) Decreased expression of the slow type I skeletal muscle fibers myosin heavy chain beta (MYH7) in those with PCOS (n=10/group), differences are based on the limma method and presented as mean ± SD. (**C**) Lower expression of proteins in slow-twitch type I muscle fibers in PCOS vs controls (p<0.05, $\log_2$ fold change <-0.5). (**D, E**) Immunofluorescent staining of type I muscle fibers with myosin heavy chain beta, and (**E**) the cell membrane with WGA. (**F**) Quantification of fiber cross-sectional area (CSA) in (**E**), difference is based on Mann-Whitney U test and data are presented as mean ± SD. (**G**) Differently phosphorylated sites in proteins expressed in muscle filaments (p<0.05, $\log_2$ fold change ± 0.5).

FC ≥0.5 or ≤–0.5, **Figure 2A**, ). 25 proteins were upregulated and 33 were downregulated in women with PCOS and the $\log_2$ fold change in expression ranged from –3.06 to 1.21 (**Supplementary file 1a**). We searched for enriched signaling pathways among the differently expressed proteins using STRING analysis. Our network had significantly more interactions than expected (enrichment p-value <1e⁻¹⁶). This means that the differently expressed proteins have more interactions with each other than what would be expected from a random set of proteins of the same size. Such an enrichment indicates that the proteins are biologically linked. Upregulated proteins were enriched in lipid metabolic pathways including negative regulation of cholesterol transport, regulation of lipoprotein lipase activity, and negative regulation of metabolic processes (**Supplementary file 1b**). This enrichment was driven by increased expression of apolipoproteins C-I, C-II, and C-III (**Figure 2B**), which are also enriched in the negative regulation of lipoprotein lipase activity (GO:0051005). Aldo-keto reductase family 1 members C1 and C3 (*AKR1C1* and *AKR1C3*, **Figure 2B**), which have an androsterone dehydrogenase activity (GO:0047023), were also upregulated and AKR1C1 was strongly correlated to higher circulating testosterone levels (Spearman's rho = 0.65, p=0.002), suggesting that muscle may produce testosterone via the backdoor pathway. Moreover, perilipin-1 that typically coats the surface of lipid droplets in adipocytes (**Gandolfi et al., 2011**; **Zhao et al., 2021**), the so-called extra-myocellular adipocytes, was increased in PCOS muscle. The increased expression of perilipin-1 was confirmed by immunofluorescence staining and quantification of muscle biopsies (**Figure 2C and D**).

The downregulated proteins in PCOS were enriched in pathways involved in muscle contraction, actin filament organization, and transition between fast and slow fibers (**Figure 3A**). All significantly enriched pathways are listed in **Supplementary file 1b**. Expression of myosin heavy chain beta, which is specific for type I muscle fibers was decreased in PCOS (**Figure 3B**). Several proteins that are more highly expressed in type I muscle fibers consistently had a lower expression in women with PCOS, e.g., myosin heavy chain 7, myosin regulatory light chains 2 and 3, troponin I and troponin T in slow

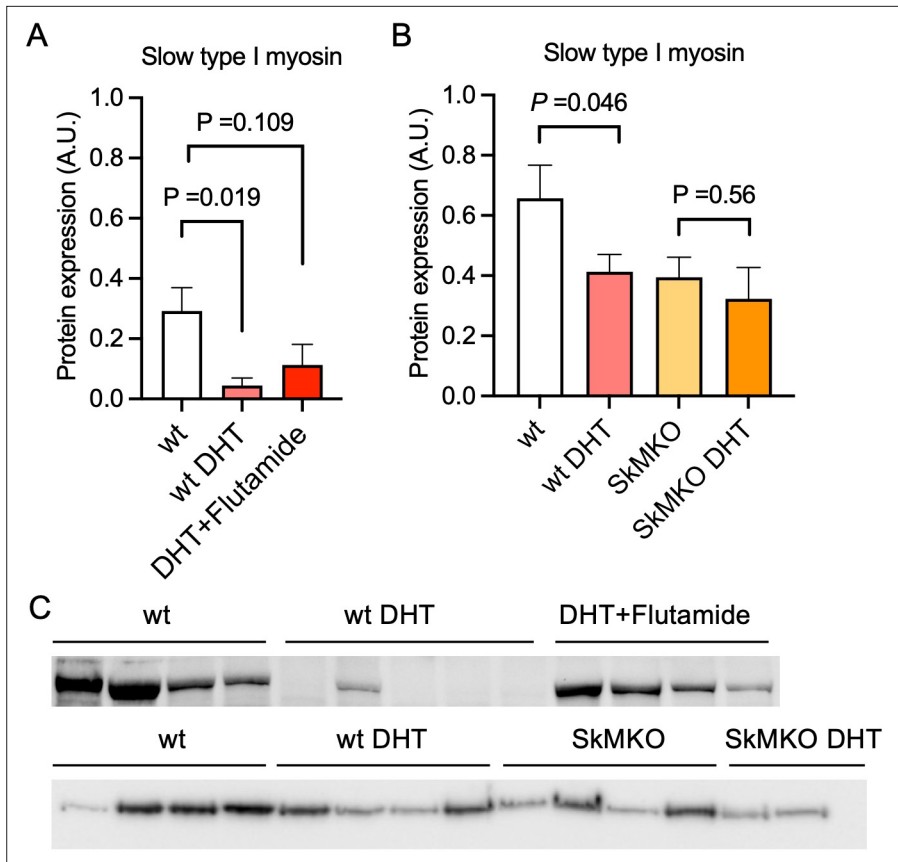

**Figure 4.** Androgen exposure leads to a shift in muscle fiber type in mice. (**A**) Decreased expression of the slow type I skeletal muscle fibers myosin heavy chain beta (MYH7) in dihydrotestosterone (DHT)-exposed polycystic ovary syndrome (PCOS)-like mice. This effect was partly blocked by the androgen receptor antagonist flutamide (n=5–6/group). (**B**) Decreased expression of slow type I skeletal muscle fibers (MYH7) in skeletal muscle-specific androgen receptor knockout mice (SkMARKO) compared to wild type (wt) (p=0.033). DHT exposure did not alter the number of type I fibers in SkMARKO (n=6–8/group). (**C**) Representative expression of myosin heavy chain beta. Differences in (**A**) are based on one-way ANOVA with Dunnets multiple comparisons test and (**B**) on two-way ANOVA, and presented as mean ± SEM. The full raw unedited uncropped blots with the relevant bands clearly labeled are provided as **Figure 4—source data 1**.

The online version of this article includes the following source data for figure 4:

**Source data 1.** Contains the raw unedited uncropped blots used to generate the figure.

skeletal muscle, and the Ca$^{2+}$ pump sarcoendoplasmic reticulum calcium ATPase 2 (SERCA2a). These are proteins located in both the thick filaments (myosin regulatory light chains) and the thin filaments (troponins) of slow-twitch fibers (**Figure 3C**). A decrease in type I slow-twitch muscle fibers was also supported by staining of human myosin heavy chain 7 (MYH7) as a marker (**Figure 3D**). To further assess whether there was a reduced fiber size or decreased number of type I fibers, fiber cross-sectional area of the fibers and the percentage of type I fibers were analyzed. The quality of muscle biopsies was impaired in the PCOS group; therefore, 60% fewer fibers from each individual were analyzed in the PCOS group compared with controls (p=0.02). There was no significant difference in the mean cross-sectional area of the fibers (4530±720 in controls versus 4281±902 µm in PCOS, p=0.64, n=5–7/group) or the percentage of type I fibers (48±12 vs 45±17 µm, p=0.69, n=4/group) in the relatively few individuals analyzed (**Figure 3E**). There was, however, a decrease in the individual fiber cross-sectional area in PCOS muscle versus controls (**Figure 3F**).

Then, an androgen-exposed PCOS-like mouse model was used to corroborate that androgen exposure leads to a shift in muscle fiber type. These PCOS-like mice have longer anogenital distance, are in a chronic diestrus phase, and have glucose intolerance (**Xiong et al., 2022**; **Ascani et al., 2023**). These effects are not present in DHT-exposed mice receiving the androgen receptor antagonist

flutamide (*Ascani et al., 2023*). DHT-exposed PCOS mice had fewer type I muscle fibers compared to controls (*Figure 4A–C*). This effect was partly prevented in DHT-exposed mice receiving flutamide, supporting an effect of androgen receptor activation on muscle fiber type (*Figure 4A and C*). However, although flutamide treatment improves glucose sensitivity in PCOS-like mice, insulin resistance likely also contributes to loss in type I fibers. Moreover, DHT-exposed SkMARKO mice were used to further investigate the contribution of androgen receptor-mediated actions in skeletal muscle. While unchallenged SkMARKO mice had fewer type I muscle fibers compared to wt mice (p=0.033), they were protected against the androgen-induced type I muscle fiber loss (*Figure 4B and C*). These data suggest that androgens have direct effects on shifting the muscle fibers toward fewer type I fibers in adult females, which can be prevented by precluding signaling through androgen receptors.

We searched for overlap between the differentially expressed proteins in skeletal muscle in this study and the differentially expressed genes in our previous meta-analysis of gene expression array data (*Manti et al., 2020*). As suspected, the overlap between gene expression and protein levels was small. We found that 1 upregulated and 12 downregulated genes in muscle biopsies from women with PCOS were also differently expressed at the protein level in this study (*Supplementary file 1c*). Several proteins involved in skeletal muscle contraction were consistently downregulated at the mRNA expression level in muscle tissue from women with PCOS including *MYL3, MYOZ2, TNNT1, LMOD2, NRAP*, and *XIRP1* (*Supplementary file 1c*).

We identified 5512 phosphosites in muscle, and 61 sites in 40 unique proteins were differentially phosphorylated in PCOS versus controls (*Supplementary file 1d*), suggesting different protein activity. Eleven of the differently expressed proteins had one or more differently phosphorylated sites, including increased phosphorylation of Ser 130, 382, and 497 in perilipin-1. Many of the proteins in the thick and thin filaments had one or more altered phosphorylation sites (*Figure 3G*). There were no significantly enriched pathways among the differently expressed phosphosites.

## Total protein expression and phosphorylation in adipose tissue

In total, we identified 5000 proteins in adipose tissue but the difference between groups was modest. 21 unique proteins were found to be differentially expressed in adipose tissue from women with PCOS versus controls (p<0.05, and $\log_2$ FC ≥0.5 or ≤–0.5, *Figure 5A*). Six proteins were upregulated and 15 were downregulated in women with PCOS and the $\log_2$ fold change in expression ranged from 2.1 to –1.6 (*Figure 5B*, *Supplementary file 1c*). Several of the upregulated proteins play a role in immune system processes including immunoglobulins, human leukocyte antigen (HLA) class I histocompatibility antigen, and sequestosome 1. Sequestosome 1 may also regulate the mitochondrial organization (*Poon et al., 2021*). Three mitochondrial matrix proteins – tRNA pseudouridine synthase A, Enoyl-CoA hydratase domain-containing protein 2, and NAD kinase 2 – had an altered expression (*Figure 5B*), possibly indicating mitochondrial dysfunction. There were three significantly enriched signaling pathways in adipose tissue based on a lower expression of leiomodin-1 and adseverin: actin nucleation (GO:0045010), positive regulation of cytoskeleton organization (GO:0051495), and positive regulation of supramolecular fiber organization (GO:1902905).

Both low-grade inflammation and transforming growth factor beta (TGFβ)-induced fibrosis have been suggested to play a role in the pathophysiology of PCOS (*Mancini et al., 2021*; *McIlvenna et al., 2021*). Dysregulated signaling of TGFβ has been linked to the development of ovarian fibrosis and reproductive dysfunction (*Stepto et al., 2019*; *McIlvenna et al., 2021*). Women with PCOS have elevated levels of circulating TGFβ1 (*Raja-Khan et al., 2010*), which is thought to trigger increased fibrotic mechanisms in other peripheral tissues. However, we did not detect an increased abundance of fibrous collagens in adipose tissue and the fibrillar collagen levels as judged by picrosirius red staining were similar between groups although the variability was higher in the PCOS group (*Figure 5C*).

We identified 5734 phosphosites in adipose tissue, of which 39 were differently phosphorylated sites in 34 unique proteins. Ten of these sites had lower phosphorylation (*Supplementary file 1f*). There was no overlap between differently phosphorylated proteins and differently expressed proteins. Perilipin-1 had two phosphorylation sites with higher phosphorylation: Ser 497 and 516 in PCOS adipose tissue compared to controls (*Supplementary file 1f*). Ser 497 showed increased phosphorylation in perilipin-1 in both muscle and adipose tissue. Under adrenergic stimulation, perilipin-1 is phosphorylated at Ser 497 by protein kinase A, which in turn triggers lipolysis by hormone sensitive lipase in adipocytes (*Marcinkiewicz et al., 2006*).

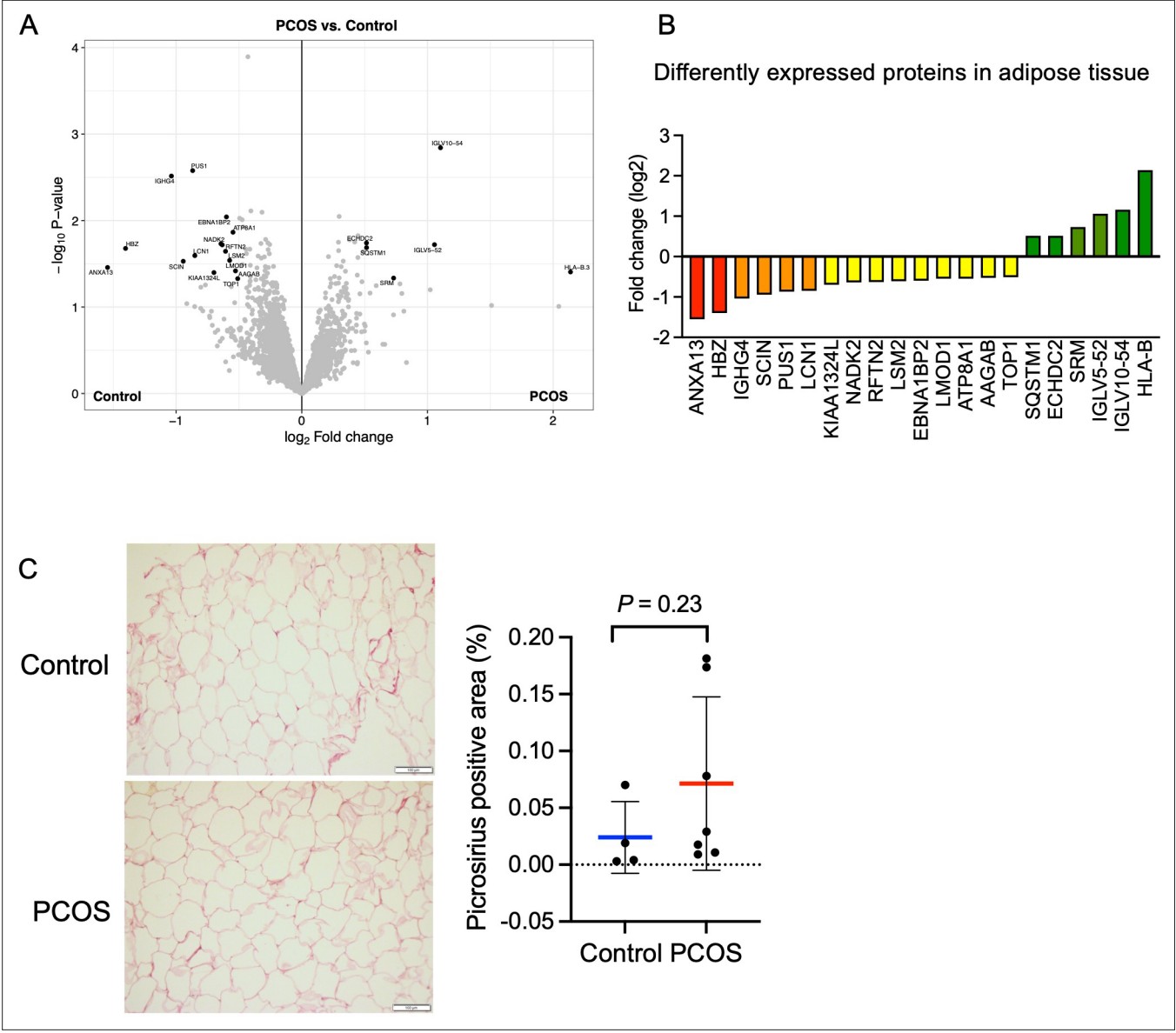

**Figure 5.** Protein expression and differently expressed proteins in adipose tissue. (**A**) Volcano plot showing the mean protein $\log_2$ fold change in adipose tissue (polycystic ovary syndrome [PCOS] vs controls) using limma method, and plotted against the $-\log_{10}$ p-value highlighting significantly regulated proteins (black; p<0.05, $\log_2$ fold change ± 0.5). n=10/group. (**B**) All differentially expressed proteins in adipose tissue from women with PCOS. (**C**) Picrosirius red staining of s.c. adipose tissue. The difference between women with PCOS (n=7) and controls (n=4) was based on Mann-Whitney U test and is presented as mean ± SD.

## Skeletal muscle protein expression and phosphorylation changes in skeletal muscle after treatment with electrical stimulation

Since long-term electrically stimulated muscle contractions improve glucose regulation and lower androgen levels in women with PCOS (*Stener-Victorin et al., 2016*), we analyzed genome-wide mRNA expression from women with PCOS after treatment to identify changes in skeletal muscle gene expression in response to stimulation. None of the transcripts exhibited changes in expression after FDR correction after 5 weeks of treatment (q<0.05), but 12 transcripts had an FC >1.2 (p<0.05, ), of which 5 were different collagens. We also analyzed whether the response to electrical stimulation was associated with DNA methylation changes in skeletal muscle. We found that 41,186 (13.8%) of 298,332 analyzed CpG sites had differential methylation in skeletal muscle after treatment (p<0.05), which is almost three times more than expected by chance (p<0.0001, $\chi^2$ test). Of these, 43 CpG sites

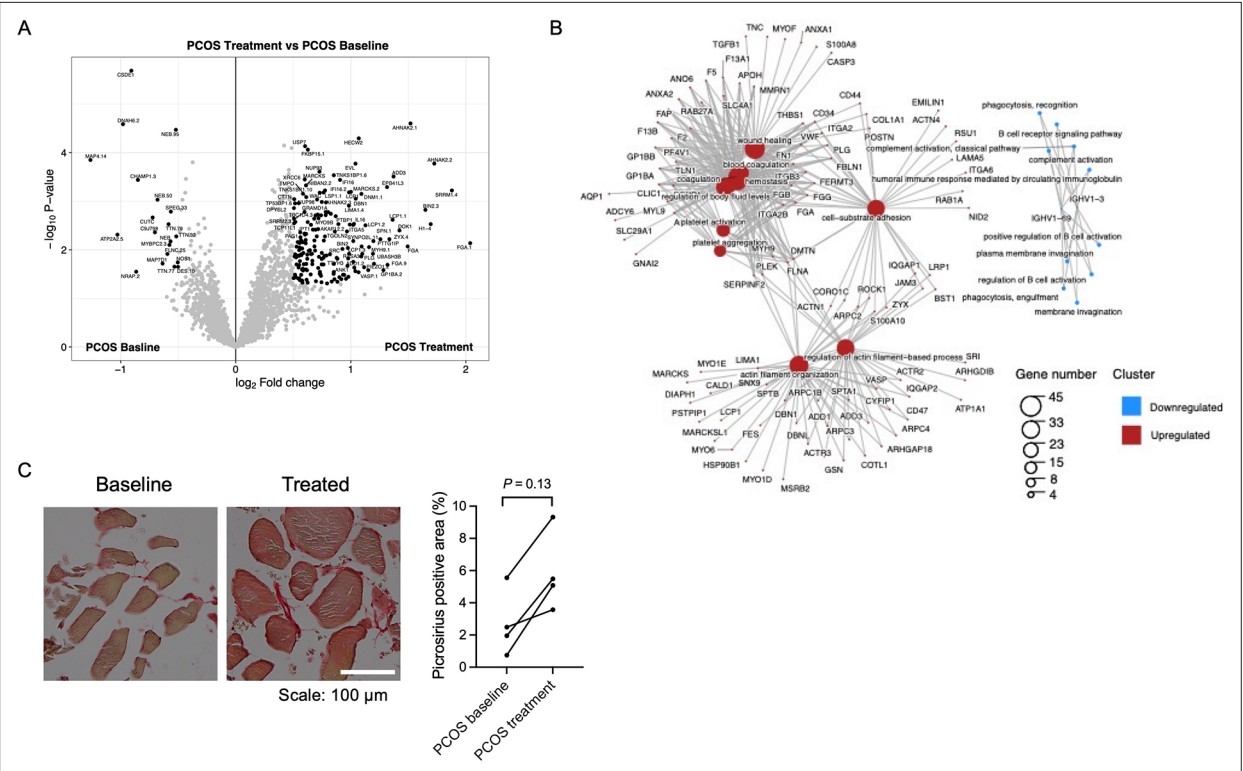

**Figure 6.** Protein expression and enriched signaling pathways in skeletal muscle after treatment with electrical stimulation. (**A**) Volcano plot showing the mean protein log$_2$ fold change in skeletal muscle (treatment vs baseline in polycystic ovary syndrome [PCOS]) using limma method, and plotted against the –log$_{10}$ p-value highlighting significantly regulated proteins (black; p<0.05, log$_2$ fold change ± 0.5). n=10/group. (**B**) GO terms for biological function of the changed proteins. (**C**) Representative pictures and quantification of picrosirius red staining of skeletal muscle before and after treatment with electrical stimulation in the same individual (n=4). Change between baseline and after treatment was based on Wilcoxon signed-rank test.

remained significant after FDR correction (q<0.05, ). The majority of the sites (74%) showed decreased methylation in response to treatment. The absolute change in DNA methylation in response to treatment ranged from 3% to 14% points.

Since mRNA expression was not significantly regulated in response to repeated electrical stimulations, we investigated whether the effects were regulated at the protein level. We found that 376 unique proteins were changed in skeletal muscle after treatment with electrical stimulation (p<0.05, *Figure 6A*, *Supplementary file 1h*). Most proteins were upregulated in women with PCOS after treatment (98%), and the log$_2$ fold change in expression ranged from –1.37 to 2.15. The upregulated proteins were enriched in signaling pathways involved in extracellular matrix (ECM) organization, regulation of TGFβ production, neutrophil-mediated immunity, wound healing, and blood coagulation (*Figure 6B*, *Supplementary file 1i*). Proteins involved in ECM organization include eight different collagens, integrins, and TGFβ1 (*Supplementary file 1i*). Collagen 1A1, 1A2 and VCAN were increased after treatment at both gene and protein levels, implying that acupuncture needling elicits a wound healing response. There was a trend toward increased staining of fibrous collagen after treatment, but this was not significant, potentially due to the low number of analyzed good quality sections (*Figure 6C*). Other upregulated signaling pathways include exocytosis and vesicle transport along the actin filament, muscle contraction, and actin filament organization and negative regulation of adenylate cyclase-activating adrenergic signaling. Several effects previously shown to be upregulated after one bout of electroacupuncture were regulated in the opposite direction as pathways involved in negative regulation of angiogenesis, negative regulation of blood vessel morphogenesis, negative regulation of nitric oxide metabolic processes, and regulation of vasoconstriction were enriched. None of the 58 proteins that had a different expression in the PCOS group at baseline were reversed after 5 weeks of treatment.

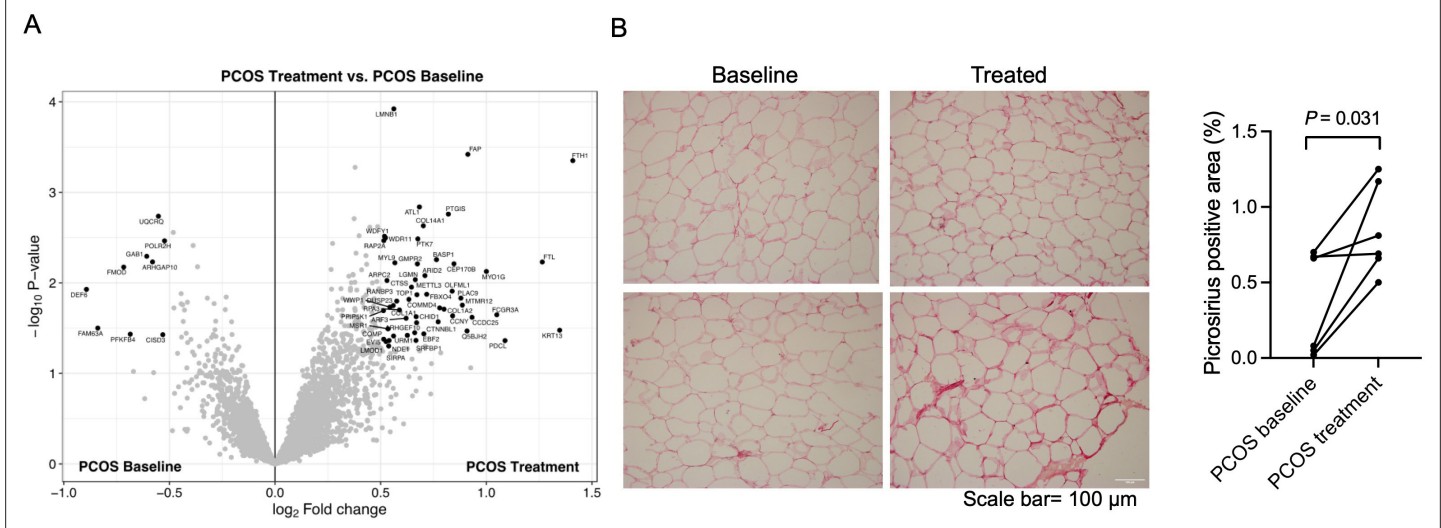

**Figure 7.** Protein expression and collagen quantification in adipose tissue after treatment with electrical stimulation. (**A**) Volcano plot showing the mean protein $log_2$ fold change in adipose tissue (treatment vs baseline in polycystic ovary syndrome [PCOS]) using limma method, and plotted against the $-log_{10}$ p-value highlighting significantly regulated proteins (black; p<0.05, $log_2$ fold change ± 0.5). n=10/group. (**B**) Representative pictures and quantification of picrosirius red staining of adipose tissue before and after treatment with electrical stimulation (n=6). Changes between baseline and after treatment were based on Wilcoxon signed-rank test.

198 phosphosites in 152 unique proteins showed a changed phosphorylation in response to electrical stimulation. 178 sites had higher phosphorylation, and 46 sites were less phosphorylated (**Supplementary file 1j**). There were no significant enriched pathways among the proteins with differently expressed phosphorylation sites. 38 of the differently expressed proteins had one or more differently phosphorylated sites. These proteins, with changes in both total protein and phosphorylation levels, were enriched in actin filament organization (GO:0007015).

## Total protein expression and phosphorylation in adipose tissue after treatment

Similar to skeletal muscle, long-term electrical stimulations had minimal effects on gene expression in adipose tissue. None of the transcripts exhibited changes in expression after FDR correction (q<0.05) after 5 weeks of treatment, or had a FC >1.2 (data not shown). We found that 23,517 (7.9%) of 298,289 analyzed CpG sites had differential methylation in adipose tissue after 5 weeks of treatment (p<0.05), which is more than expected by chance (p<0.0001, $\chi^2$ test). The majority (63.5%) of these sites showed reduced methylation in response to treatment. One CpG site remained significant after FDR correction (q<0.05, –2.2% points reduced methylation of cg13383058 in the transcription start site of *CD248*). Therefore, we investigated whether the long-term effects were regulated at the protein level. 61 unique proteins were found to be changed in adipose tissue after electrical stimulation treatment (**Figure 7A**, **Supplementary file 1k**). Most of the proteins were upregulated (85%) and nine were downregulated in women with PCOS after treatment, and the $log_2$ fold change in expression ranged from –0.89 o 1.41. The upregulated enriched signaling pathways include ECM organization and Fc-gamma receptor signaling (**Supplementary file 1l**). In accordance with these findings, 5 weeks of treatment increased the fibrillar collagen content in adipose tissue (**Figure 7B**). The expression of DNA topoisomerase 1 and leimodulin-1 was lower in women with PCOS than in controls but increased after treatment (**Supplementary file 1k**). Leimodulin-1 is required for proper contractility of smooth muscle cells by mediating nucleation of actin filaments, and myosin regulatory light polypeptide 9 plays an important role in regulating smooth muscle contractile activity. The enzyme prostacyclin synthase, a potent mediator of vasodilation, was also increased and could act on the smooth muscles in the vessel wall. 49 phosphosites in 46 unique proteins showed altered phosphorylation in response to electrical stimulation. All except four sites had higher phosphorylation (**Supplementary file 1m**). There were no significantly enriched pathways among the proteins with differently expressed phosphosites.

11 proteins showed higher expression in both skeletal muscle and adipose tissue after treatment: FCGR3A, FAP, PTGIS, COL1A2, COL14A1, COL1A1, MYL9, LMNB1, SIRPA, ARPC2, and RAP2A. These proteins are enriched in ECM organization (GO:0030198), negative regulation of nitric oxide biosynthetic/metabolic processes (GO:0045019, GO:1904406), and Fc-gamma receptor signaling pathways (GO:0038096, GO:0038094, GO:0002431).

## Discussion

### Proteome signature in PCOS

We have profiled the proteome of skeletal muscle and adipose tissue to advance our understanding of the pathophysiology of PCOS. The changes in protein expression in adipose tissue were small, whereas in skeletal muscle of women with PCOS there was a clear downregulation of proteins involved in muscle contraction. The skeletal muscle contains a mixture of slow-twitch oxidative and fast-twitch glycolytic myofibers, which exhibit different physiological properties. Type I, or red fibers, are slow-twitch fatigue-resistant muscle fibers that have higher mitochondrial and myoglobin content, and are thus more aerobic than type II fast-twitch fibers. Several proteins specific to or known to be highly expressed in type I muscle fibers were consistently downregulated in women with PCOS. These proteins are located in both the thick and the thin filaments of slow-twitch fibers. These data suggest that there are fewer type I fibers, decreased number and/or smaller, in the PCOS muscle. Unfortunately, we were unable to quantify the number of type I fibers in the entire cohort because of the poor quality of the PCOS muscle biopsies and the unavailability of muscle tissue. We also identified several differently phosphorylated sites in proteins located in the thick and thin filaments, indicating a differential protein activity, since phosphorylation can change the activity of a protein.

Here, we show that the signaling pathway important for the transition between fast and slow fibers was downregulated and that individuals with PCOS had lower expression of myosin heavy chain beta (encoded by *MYH7*), which is specific for slow-twitch oxidative type I fibers (*Schiaffino and Reggiani, 2011*). A decrease in type I fibers has been shown in three different androgen-excess rodent models, including this study (*Holmäng et al., 1990*; *Shen et al., 2019*). This effect on type I fibers was partly protected by the coadministration of the androgen receptor antagonist flutamide. Moreover, mice that lack the AR specifically in skeletal muscle were completely protected against this effect. These findings suggest that exaggerated androgen signaling in skeletal muscle directly affects the muscle fiber-type composition. While we found a lower abundance of type I muscle fiber in muscle-specific AR knockout females, a recent study shows that depleting AR signaling in skeletal muscle does not lead to necrotic fibers, aberrant histology, or changes in key metabolic functions in females (*Ghaibour et al., 2023*). Thus, androgen signaling is likely to be important for normal muscle development but may play a different and dose-dependent role in adulthood. Impaired insulin sensitivity in hyperandrogenic animals is associated with fewer type I fibers and increased type II fibers in skeletal muscles (*Holmäng et al., 1990*; *Shen et al., 2019*), features expected to result in reduced insulin sensitivity in this tissue as a higher proportion of oxidative type I fibers leads to better insulin responsiveness (*Stuart et al., 2013*). In line with a higher mitochondrial content in type I fibers, there was a lower expression of several mitochondrial matrix proteins: enoyl acyl carrier protein reductase, 3-oxoacid CoA-transferase 1, enoyl-CoA delta isomerase 1, hydroxyacid-oxoacid transhydrogenase, and acetyl-CoA synthetase 2 (GO:0005759, q=0.004) in women with PCOS. Moreover, a lower expression of the mitochondrial acetyl-CoA synthetase 2 correlated with a higher HOMA-IR (Spearman's rho = −0.46, p=0.04), suggesting that an impaired mitochondrial function contributes to insulin resistance. Moreover, a lower proportion of type I fibers correlated with the severity of insulin resistance in subjects with the metabolic syndrome (*Stuart et al., 2013*). Androgens appear to decrease highly oxidative and insulin-sensitive type I muscle fibers and increase glycolytic and less insulin-sensitive type II fibers in non-athletes, further promoting the development of insulin resistance. However, whether these changes in muscle morphology precede or follow the development of insulin resistance in women with PCOS is not known. Moreover, the SERCA2a, which is expressed primarily in slow-twitch skeletal muscle, was less abundant in PCOS muscle. The SERCA2a pump transports calcium ions from the cytosol back to the sarcoplasmic reticulum after muscle contraction to keep the cytosolic $Ca^{2+}$ concentration at a low level. Its function is closely related to muscle health and function (*Xu and Van Remmen, 2021*). There were three sites with lower phosphorylation in SERCA2a,

suggesting that lower SERCA2a expression is not only a reflection of decreased type I fibers but also an altered function. Impaired SERCA2 function may lead to increased cytosolic $Ca^{2+}$ concentration, which in turn can impair force production and mitochondrial function in type I fibers (*Xu and Van Remmen, 2021*). Although serum androgen levels are positively correlated with athletic performance in female athletes (*Bermon et al., 2018*), skeletal muscle contraction and filament sliding pathways were downregulated in muscle biopsies from hyperandrogenic women with overweight/obesity in this study. Thus, androgens may have differential actions on female skeletal muscle function in moderately physically active subjects and female athletes.

Androgens are mainly produced in the ovaries and adrenal glands but there is also a local production in adipose tissue. In overweight/obese women with PCOS, increased AKR1C3 levels mediated increased testosterone generation from androstenedione in subcutaneous adipose tissue, which enhanced lipid storage (*O'Reilly et al., 2017*). Higher expression of AKR1C1 and AKR1C3 in PCOS skeletal muscle in this study could increase local synthesis of androgens via the backdoor pathway, and increase androgenic signaling in skeletal muscle.

The expression of proteins involved in lipid transport and negative regulation of lipid metabolism was increased in the muscles of women with PCOS. Lipid transport was clustered around apolipoproteins C1, C2, and C3. Interestingly, a recent proteomics study shows that serum apolipoprotein C3 levels are higher in insulin-resistant women with PCOS compared to insulin-sensitive women (*Li et al., 2020*). Could apolipoproteins aid in the deposition of excess fat in skeletal muscle and contribute to lipotoxicity? Fatty acids in skeletal muscle that do not undergo beta-oxidation in the mitochondria inevitably contribute to lipid synthesis. There was indeed a lower expression of various mitochondrial proteins involved in mitochondrial fatty acid beta-oxidation, enoyl acyl carrier protein reductase, enoyl-CoA delta isomerase 1, and acyl-CoA thioesterase 11 (R-HSA-77289, q=0.0008) in PCOS muscle. The most important single fate for these fatty acids is triacylglycerol esterification and storage in lipid droplets. Perilipins act largely as a scaffold protein on lipid droplets. Perilipin-1 is localized in the periphery of intramuscular lipids, known as extracellular lipids (*Gandolfi et al., 2011*; *Zhao et al., 2021*). In pig, muscle perilipin-1 proteins are localized around lipid droplets in mature and developing adipocytes in correspondence with extra-myocellular lipids (*Gandolfi et al., 2011*; *Zhao et al., 2021*). Therefore, the high expression of peilipin-1 in muscle from women with PCOS is likely a sign of lipotoxicity. Taken together, intramuscular lipid accumulation and type I muscle fiber decline likely contribute to insulin resistance in PCOS muscle.

PCOS is associated with chronically elevated levels of inflammatory markers in the circulation (*Orio et al., 2005*) and low-grade inflammation has been suggested to play a key role in the pathophysiology (*Mancini et al., 2021*). We and others have previously shown that many of the significantly enriched gene expression pathways in PCOS muscle are involved in immune responses or are related to immune diseases in muscle (*Skov et al., 2007*; *Nilsson et al., 2018*). A distinct pattern was the downregulation of a family of genes named the HLA complex and downregulation of gene sets associated with inflammatory responses in the PCOS muscle (*Manti et al., 2020*). In this study, the HLA-B was the most downregulated protein, in line with lower gene expression of HLA-B in skeletal muscle (*Nilsson et al., 2018*). The HLA complex helps the immune system distinguish endogenous proteins from proteins made by foreign invaders by making peptide-presenting proteins that are present on the cell surface. When the immune system recognizes the peptides as foreign, this will trigger self-destruction of the infected cell. Numerous HLA alleles have been identified as genetic risk factors for autoimmune thyroid disease, which is increased nearly threefold in women with PCOS (*Zeber-Lubecka and Hennig, 2021*). The co-occurrence of PCOS and autoimmune thyroid disease has led to the suggestion that PCOS itself may be an autoimmune disorder. At the same time, five different immunoglobulins were upregulated, which contributed to the enrichment of immune system pathways.

In adipose tissue, the difference between groups was small with relatively few differently expressed proteins but several of the proteins with a higher expression play a role in immune system processes including immunoglobulins and HLA-B. Sequestosome-1 is an immunometabolic protein that is both involved in the activation of nuclear factor kappa-B and regulates mitochondrial functionality by modulating the expression of genes underlying mitochondrial respiration (*Poon et al., 2021*). Alterations in the immune response and immunometabolic pathways are seen in both muscle and adipose tissue, but whether and how these alterations affect and potentially contribute to the pathophysiology of PCOS remains to be investigated.

## Electrical stimulation-related proteome changes

Regular physical exercise improves insulin sensitivity and is the first-line approach to manage both reproductive and metabolic disturbances in those with PCOS and overweight or obesity, but adherence is low. Electrical stimulations inducing contraction could therefore be an alternative way to ameliorate symptoms in women with PCOS, alongside exercise. Transcriptomic changes in response to one single bout of electrical stimulation mimic the response to one bout of exercise (*Benrick et al., 2020*). Therefore, we hypothesize that long-term treatment with electrical stimulation or exercise also triggers overlapping signaling pathways. Five weeks of repeated treatment with electrical stimulation did not alter mRNA expression but increased the expression of several hundred proteins in skeletal muscle and about 50 proteins in adipose tissue. The most pronounced changes in both tissues were mechanisms involved in wound healing such as the increase in ECM formation and enriched pathways for nitric oxide metabolic process and Fc-gamma receptor signaling pathways. Moreover, wound healing and blood coagulation pathways were upregulated in skeletal muscle, suggesting that repeated needling induces small damages in the tissues that trigger a wound healing response. However, increased expression of ECM-related structural components such as collagens and integrins are not only involved in wound healing, but are also increased in response to muscle contractions as previously shown in response to eccentric exercise and repeated electrical stimulation with electrodes (*Mackey et al., 2011*; *Hyldahl et al., 2015*). We propose that these changes are induced by contraction but we cannot rule out that the ECM-related changes occurred as a direct result of repeated needle insertion. The delayed synthesis of collagens and subsequent strengthening of the ECM structural matrix in response to both exercise and electrical stimulation without needle insertion (*Mackey et al., 2011*; *Hyldahl et al., 2015*) supports the idea that contractions induce remodeling of the ECM that may provide protective adaptation to repeated contractions.

We hypothesized that electrical stimulations mimic the response to repeated exercise. Therefore, we searched for overlap between changes in the proteome following electrical stimulation treatment in this study and a meta-analysis on the exercise-induced proteome in skeletal muscle (*Padrão et al., 2016*). Seven proteins were changed in both conditions: fibrinogen beta-chain, actin, vimentin, annexin A2, moesin, gelsolin, and hypoxanthine-guanine phosphoribosyl-transferase. Actin, vimentin, and moesin make up structural parts of the cytoskeleton and all proteins are enriched in the GO terms immune system processes (GO:0002376) and positive regulation of metabolic processes (GO:0009893). Current evidence suggests that there are immunometabolic alterations in skeletal muscle from women with PCOS (*Skov et al., 2007*; *Nilsson et al., 2018*; *Manti et al., 2020*; *Stepto et al., 2020*), and increased expression of the abovementioned proteins could cause contraction-induced changes in the immunometabolic response.

Repeated treatment with electrical stimulation improves glucose homeostasis and lowers Hba1c in women with PCOS (*Stener-Victorin et al., 2016*). A not so well-characterized, upregulated pathway involves exocytosis and protein transport, which is of interest with regard to glucose transporter 4 (GLUT4) translocation stimulated by AMP kinase (AMPK). The activity of AMPK in muscle increases substantially during contraction and increases glucose transport. AMPK has been linked to at least two mechanisms for the control of vesicle trafficking, namely the regulation of Rab and the generation of phosphatidylinositol 3,5-bisphosphate in the control of GLUT4 translocation (*Sylow et al., 2017*). The Rab GTPase-activating proteins TBC1D1 and TBC1D4 are thought to mediate the effects of AMPK on GLUT4 translocation and glucose transport. At present, it is unclear which specific Rab proteins might be regulated by AMPK downstream of TBC1D1 and TBC1D4, but Rab-13 appears to act downstream of TBC1D4 (*Sylow et al., 2017*). Six Rab proteins showed higher expression after electrical stimulation, including Rab-13, making these proteins potential candidates for regulating glucose uptake by electrically stimulated contractions. The proteome in adipose tissue and skeletal muscle did not show differential expression of proteins with well-known metabolic effects that can easily explain the improvement in Hba1c. This could be due to the timepoint when the biopsies were collected, which was within 48 hr of the last treatment when the acute effects of electrical stimulation are no longer present and the long-term changes may be more subtle.

## Limitations

The interpretation of the protein expression in this study is limited by the relatively small number of women (n=10/group), the inclusion of only Caucasian women, and a mix of PCOS phenotypes since

we used the Rotterdam diagnostic criteria. Moreover, we cannot distinguish whether the identified dysregulated pathways are responses to PCOS or causal effectors. Some may argue that another limitation is the fact that we cannot identify the changes in protein expression in specific cell types, i.e., adipocytes and myocytes, as the biopsies consist of many different cell types and structures, e.g., nerves, immune cells, vessels, and connective tissue. However, this can also be seen as a strength, as no cell acts independently of the cells surrounding it.

## Conclusions

Our findings suggest that highly oxidative and insulin-sensitive type I muscle fibers are decreased in PCOS which in combination with more extra-myocellular lipids may be key factors for insulin resistance in PCOS muscle. In adipose tissue, the difference between groups was small. A 5-week treatment with electrical stimulation triggered a wound healing response in both adipose tissue and skeletal muscle. In addition, remodeling of the ECM can provide protective adaptation to repeated skeletal muscle contractions.

## Acknowledgements

For proteomic analysis we thank Britt-Marie Olsson, Johannes Fuchs, and Carina Sihlbom at the Proteomics Core Facility, and for the use of its technical equipment and support we thank the Genomics Core Facility, both at Sahlgrenska Academy, University of Gothenburg, Sweden. We are grateful to the Inga-Britt and Arne Lundberg Research Foundation for the donation of the Orbitrap Fusion Tribrid MS instrument. For analysis of array data we thank Michaela Martis at University of Linköping. AB holds funding from the Swedish Research Council (2020-02485), ESV holds funding from the Swedish Research Council (2022-00550), the Novo Nordisk Foundation (NNF22OC0072904), and IWA holds funding from the Swedish Research Council (2020-01463), Mary von Sydow Foundation, Diabetes Wellness Sverige, and EFSD//European Research Programme on 'New Targets for Diabetes or Obesity-related Metabolic Diseases' supported by MSD 2022, and JN holds funding from IngaBritt and Arne Lundberg Research Foundation. The funding bodies did not have a role in study design and have no role in the implementation of the study.

## Additional information

### Funding

| Funder | Grant reference number | Author |
| --- | --- | --- |
| Vetenskapsrådet | 2020-02485 | Anna Benrick |
| Vetenskapsrådet | 2022-00550 | Elisabet Stener-Victorin |
| Novo Nordisk Fonden | NNF22OC0072904 | Elisabet Stener-Victorin |
| Vetenskapsrådet | 2020-01463 | Ingrid Wernstedt Asterholm |
| Stiftelsen Mary von Sydows, född Wijk, donationsfond | | Ingrid Wernstedt Asterholm |
| Insamlingsstiftelsen Diabetes Wellness Network Sverige | | Ingrid Wernstedt Asterholm |
| IngaBritt och Arne Lundbergs Forskningsstiftelse | | Jenny Nyström |

The funders had no role in study design, data collection and interpretation, or the decision to submit the work for publication.

## Author contributions

Elisabet Stener-Victorin, Conceptualization, Supervision, Funding acquisition, Investigation, Project administration, Writing – review and editing; Gustaw Eriksson, Data curation, Formal analysis, Methodology, Writing – review and editing; Man Mohan Shrestha, Valentina Rodriguez Paris, Haojiang Lu, Jasmine Banks, Manisha Samad, Charlène Perian, Baptiste Jude, Viktor Engman, Roberto Boi, Data curation, Formal analysis; Emma Nilsson, Data curation, Formal analysis, Writing – original draft; Charlotte Ling, Nigel Turner, Johanna Lanner, Supervision, Methodology; Jenny Nyström, Resources, Methodology; Ingrid Wernstedt Asterholm, Supervision, Methodology, Writing – review and editing; Anna Benrick, Conceptualization, Data curation, Formal analysis, Funding acquisition, Visualization, Writing – original draft, Project administration, Writing – review and editing

## Author ORCIDs

Elisabet Stener-Victorin ⓘ https://orcid.org/0000-0002-3424-1502
Gustaw Eriksson ⓘ http://orcid.org/0000-0003-0120-9028
Emma Nilsson ⓘ https://orcid.org/0000-0001-5020-6582
Ingrid Wernstedt Asterholm ⓘ http://orcid.org/0000-0002-0755-5784
Anna Benrick ⓘ https://orcid.org/0000-0003-4616-6789

## Ethics

Clinical trial registration The study was registered at ClinicalTrials.gov (NTC01457209).

The study was conducted at the Sahlgrenska University Hospital and the Sahlgrenska Academy, University of Gothenburg, Gothenburg, Sweden, in accordance with the standards set by the Declaration of Helsinki. Procedures have been approved by the Regional Ethical Review Board of the University of Gothenburg (approval number 520-11). All women provided oral and written informed consent before participation in the study.

All animal experiments were carried out in compliance with the ARRIVE guidelines. Procedures were approved by the Sydney Local Health District Animal Welfare Committee within National Health and Medical Research Council guidelines for animal experimentation or the Stockholm Ethical Committee for Animal Research (approval number 20485-2020).

Reviewer #1 (Public Review): https://doi.org/10.7554/eLife.87592.3.sa1
Reviewer #2 (Public Review): https://doi.org/10.7554/eLife.87592.3.sa2
Author Response https://doi.org/10.7554/eLife.87592.3.sa3

---

# Additional files

## Supplementary files

- MDAR checklist

- Supplementary file 1. Differently expressed proteins, phosphorylation sites, and transcripts, and their respective enriched pathways, in skeletal muscle and adipose tissues. (a) Differentially expressed proteins in skeletal muscle from women with polycystic ovary syndrome (PCOS) compared with controls (n=10/group). p<0.05, $\log_2$ fold change ± 0.5. (b) Gene ontology pathway analysis of differentially expressed proteins in skeletal muscle from women with PCOS compared with controls. (c) Overlap between differentially expressed proteins in skeletal muscle from women with PCOS compared with controls and differentially expressed genes. (d) Phosphorylation sites with a change in phosphorylation in skeletal muscle from women with PCOS compared with controls. (e) Differentially expressed proteins in adipose tissue from women with PCOS compared with controls. (f) Phosphorylation sites with a change in phosphorylation in adipose tissue from women with PCOS compared with controls. (g) Gene expression and methylation changes in skeletal muscle after treatment with electrical stimulation. (h) Changed proteins in skeletal muscle from women with PCOS after treatment with electrical stimulation. (i) Gene ontology pathway analysis of changed proteins in skeletal muscle from women with PCOS after treatment with electrical stimulation. (j) Phosphorylation sites with a change in phosphorylation in skeletal muscle from women with PCOS after treatment with electrical stimulation. (k) Changed proteins in adipose tissue from women with PCOS after treatment with electrical stimulation. (l) Gene ontology pathway analysis of changed proteins in adipose tissue from women with PCOS after treatment with electrical stimulation. (m) Phosphorylation sites with a change in phosphorylation in adipose tissue from women with PCOS

after treatment with electrical stimulation.

## Data availability

The study was registered at ClinicalTrials.gov (NTC01457209). The mass spectrometry proteomics data have been deposited to the ProteomeXchange Consortium via the Proteomics Identifications (PRIDE) (RRID:SCR_003411SCR_003411) partner repository with the dataset identifier PXD025358. The protein expression analysis is published at https://github.com/GustawEriksson/FAT-MUS-Proteomics, (copy archived at *Eriksson, 2023*). Individual-level methylation and mRNA expression data are not publicly available due to ethical and legal restrictions related to the Swedish Biobanks in Medical Care Act, the Personal Data Act and European Union's General Data Protection Regulation and Data Protection Act. All other data generated or analyzed during this study are included in the manuscript and supporting files; a Source Data file provided for *Figure 4—source data 1* contains the raw unedited uncropped blots used to generate the figure, and raw data can be found at Dryad https://doi.org/10.5061/dryad.wwpzgmsr7.

The following datasets were generated:

| Author(s) | Year | Dataset title | Dataset URL | Database and Identifier |
| --- | --- | --- | --- | --- |
| Benrick A, Stener-Victorin E | 2023 | Proteomic changes in adipose tissue and skeletal muscle in women with PCOS | https://www.ebi.ac.uk/pride/archive/projects/PXD025358 | PRIDE, PXD025358 |
| Benrick A, Stener-Victorin E | 2023 | Type I fiber decrease and ectopic fat accumulation in skeletal muscle from women with PCOS | https://doi.org/10.5061/dryad.wwpzgmsr7 | Dryad Digital Repository, 10.5061/dryad.wwpzgmsr7 |

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
