## [Editor Report · eLife assessment]

This **important** work employed global proteomic and phosphorylation site analysis to examine adipose tissue and skeletal muscle samples collected at baseline from a sample of 10 women, including those with and without PCOS, both before and after 5 weeks of electrical stimulation treatment. This work significantly enhances our knowledge by demonstrating that women with PCOS who exhibit protein hyperandrogenicity have elevated extramyocellular lipid levels and a decreased number of oxidative insulin-sensitive type I muscle fibers. The **convincing** evidence supporting these conclusions makes this research of broad interest not only to scientists but also to clinicians.

---

## [Referee Report · Reviewer #1 (Public Review)]

In the manuscript, the authors tried to explore the molecular alterations of adipose tissue and skeletal muscle in PCOS by global proteomic and phosphorylation site analysis. In the study, the samples are valuable, while there are no repeats for MS and there are no functional studies for the indicted proteins, phosphorylation sites. The authors achieved their aims to some extent, but not enough.

---

## [Referee Report · Reviewer #2 (Public Review)]

This study provides the proteomic and phosphoproteomics data for our understanding of the molecular alterations in adipose tissue and skeletal muscle from women with PCOS. This work is useful for understanding of the characteristics of PCOS, as it may provide potential targets and strategies for the future treatment of PCOS. While the manuscript presents interesting findings on omics and phenotypic research, the lack of in-depth mechanistic exploration limits its potential impact.

The study primarily presents findings from omics and phenotypic research, but fails to provide a thorough investigation into the underlying mechanisms driving the observed results. Without a thorough elucidation of the mechanistic underpinnings, the significance and novelty of the study are compromised.

---

## [Author Response]

The following is the authors’ response to the original reviews.

**eLife assessment**
This study offers an inventory of proteins and their phosphorylated sites that are up- and down-regulated in the adipose tissue and skeletal muscle of women with PCOS. The data were collected and analyzed using rigorous and validated methodology, making it a useful resource for identifying targets and strategies for future PCOS treatments. However, even though some of the predicted targets are compelling, further functional validation is required to ensure the accuracy of these identified targets. If confirmed, the findings of this study would be of significant interest to a wide range of readers.

Thank you very much for the opportunity to carry out some final revisions to our manuscript and for the invitation to submit a revised version of our work for further consideration in eLife. We are grateful for the very constructive and thorough feedback provided. Consequently, our manuscript has undergone revisions to address the issues raised, providing additional data from mouse models showing that androgen receptor signaling has a direct effect on muscle fiber type.

**Public Reviews:**

**Reviewer #1 (Public Review):**
In the manuscript, the authors tried to explore the molecular alterations of adipose tissue and skeletal muscle in PCOS by global proteomic and phosphorylation site analysis. In the study, the samples are valuable, while there are no repeats for MS and there are no functional studies for the indicted proteins, phosphorylation sites. The authors achieved their aims to some extent, but not enough.

Response: Indeed, the samples are valuable but given the relatively high sensitivity and specificity of the method we don’t see why repeats for MS would increase the power of the study. The number of tissue samples analyzed would however do so. Although no functional studies have been done, we do show that hyperandrogenism is associated with a shift towards fewer type I fibers in skeletal muscle. In the revised manuscript we have added data showing that androgens (dihydrotestosterone, DHT) have a direct effect on reducing the number of type I muscle fibers in a PCOS-like mouse model. Prepubertal DHT exposure led to a dramatic decrease in type I fibers, and this effect was partly prevented by the androgen receptor antagonist flutamide (Fig. 4A). Moreover, while skeletal muscle specific AR knockout mice presented with fewer type I muscle fibers, they were protected against the DHT-induced type I muscle fiber loss (Fig. 4B).

**Reviewer #2 (Public Review):**
This study provides the proteomic and phosphoproteomics data for our understanding of the molecular alterations in adipose tissue and skeletal muscle from women with PCOS. This work is useful for understanding of the characteristics of PCOS, as it may provide potential targets and strategies for the future treatment of PCOS. While the manuscript presents interesting findings on omics and phenotypic research, the lack of in-depth mechanistic exploration limits its potential impact.The study primarily presents findings from omics and phenotypic research, but fails to provide a thorough investigation into the underlying mechanisms driving the observed results. Without a thorough elucidation of the mechanistic underpinnings, the significance and novelty of the study are compromised.

Response: We do provide solid evidence that women with PCOS have a lower expression of proteins specific for type I muscle fibers. A comprehensive exploration of the mechanism driving the observed results is not within the scope of this paper. However, we have included experimental data from a PCOS-like mouse model to strengthen our results that hyperandrogenism has a direct effect on lowering the number of type I fibers. Prepubertal dihydrotestosterone (DHT) exposure led to a dramatic decrease in type I fibers, and this effect was abolished in DHT-exposed mice with skeletal muscle-specific deletion of the androgen receptor (Fig. 4B). Moreover, the decrease in type I fibers was partly prevented by the androgen receptor antagonist flutamide in wild-type mice (Fig. 4A). Notably, unchallenged skeletal muscle specific AR knockout mice had fewer type I muscle fiber.These data indicate that muscle AR signaling is important for normal muscle development, but that exaggerated muscle AR signaling leads to decreased abundance of type I muscle fibers in adult females.

**Reviewer #1 (Recommendations For The Authors):**
1. For participant recruitment the age should be considered.

Response: The age of the women is shown in Table 1, the mean age was around 30 years. Cases and controls were matched for age, weight, and BMI at recruitment.

1. The current method is that biopsies from 10 participants are collected as a sample, biopsy from 1 participant for MS and comprehensive analysis in the group may be better.

Response: The skeletal muscle biopsies from the 10 controls and 10 women with PCOS at baseline and after 5 weeks of treatment were collected and analyzed as individual samples. For MS each sample was handled as individual samples with subsequent comprehensive analysis of each group. This has now been further clarified in the methods; paragraph Proteomic sample preparation and LC-MS/MS analysis.

1. Figure 2C, it is not convincing that "The increased expression of perilipin-1 was confirmed by immunofluorescence staining of muscle biopsies".

Response: we have quantified perilipin-1 staining in skeletal muscle cells from control and PCOS using ImageJ software (National Institutes of Health, Bethesda, MD, USA). The channels of the images were split and converted into 8-bit. The minimum and maximum thresholds were adjusted and kept constant for all the images. Regions of interest were drawn around the cells and empty space for background intensity measurement. The mean perilipin-1 intensity was measured and corrected by deducting the background. A total of 28 PCOS and 33 control cells were quantified. The quantification of perilipin-1 staining is included in Fig. 2D. Perilipin-1 staining was more abundant in skeletal muscle cells from women with PCOS.

1. Figs.3F,4C,5C,6B, methods for the quantification are needed respectively.

Response: For each of the graphs, a detailed description of how the stainings were quantified has been included in the Methods section; Histological analyses and immunofluorescence.

Fig.3F; Fiber cross-sectional area was automatically determined using MyoVision v1.0 and the proportion of type I fibers was manually counted on ImageJ. A total of 579 fibers from seven controls (60-150 fibers per muscle section) and 177 fibers (15-80 fibers per muscle section) from women with PCOS were quantified. Data are expressed as mean ± SD and graphically depicted with each individual fiber quantified.

Fig. 4C and 6B; Quantification of picrosirius red staining of adipose tissue before and after treatment with electrical stimulation was performed using a semi-automatic macro in ImageJ software. This macro allows for calculation of the total area (μm2) and the % of collagen staining from each area adjusting the minimum and maximum thresholds.. Three different random pictures per section (4-5 sections/subject) were taken at 10x or 20x magnification using a regular bright field microscope (Olympus BX60 & PlanApo, 20x/0.7, Olympus, Japan). All images were analyzed on ImageJ software v1.47 (National Institutes of Health, Bethesda, MD, USA) using this protocol https://imagej.nih.gov/ij/docs/examples/stained-sections/index.html with the following modification; threshold min 0, max 2.

Fig. 5C; Quantification of picrosirius red staining of skeletal muscle before and after treatment with electrical stimulation was performed using a semi-automatic macro in ImageJ software v1.47 (National Institutes of Health, Bethesda, MD, USA) using the same protocol as for adipose tissue described above. % of collagen staining was calculated on 8 – 10 images of different microscopic fields from each muscle sample.

**Reviewer #2 (Recommendations For The Authors):**
While the study presents some valuable research findings, it falls short in terms of providing a comprehensive understanding of the mechanistic basis of the observed outcomes. Further exploration and elucidation of the mechanisms involved would greatly enhance the quality and impact of the study. For example, the authors need to provide sufficient evidence to elucidate why PCOS patients exhibit changes in these proteins and phosphorylation sites, as well as how these changes may impact PCOS patients, such as whether they are related to fertility. It would be valuable to provide further mechanistic insights to enhance the scientific rigor of the study.I encourage the authors to further refine their research and resubmit the manuscript with a more robust and comprehensive exploration of the mechanistic aspects to strengthen its scientific merit.

Response: PCOS is characterized by reproductive and metabolic features. Changes in protein expression and phosphorylation sites in skeletal muscle and adipose tissue likely impact metabolic function to a larger degree than fertility. With that said, altered muscle function may affect insulin resistance and inflammation, thereby potentially aggravating reproductive status including ovulatory cyclicity and fertility potential. We found that aldo-keto reductase family 1 members C1 (AKR1C1) and C3 (AKR1C3), which for example can convert androstenedione to testosterone, had a higher expression in skeletal muscle. Expression of AKR1C1 was strongly correlated to higher circulating testosterone levels (Spearman rho=0.65, P=0.002), suggesting that muscle may produce testosterone via the backdoor pathway (added to the second paragraph of the results section). Moreover, a lower expression of the mitochondrial acetyl-CoA synthetase 2 correlated with a higher HOMA-IR (Spearman rho=-0.46, P=0.04), suggesting that an impaired mitochondrial fatty acid beta-oxidation contributes to insulin resistance. There was indeed a lower expression of various mitochondrial matrix proteins, some involved in mitochondrial fatty acid beta-oxidation; enoyl acyl carrier protein reductase; enoyl-CoA delta isomerase 1, and acyl-CoA thioesterase 11 (R-HSA-77289, q=0.0008) in PCOS muscle (this has been added to the discussion).

A comprehensive exploration of the mechanism driving these changes is not within the scope of this paper. However, we have added data from PCOS-like mice to strengthen the paper. This mouse model supports our hypothesis that androgens drive the shift towards less type I muscle fibers, an effect that can be partly reversed by blocking the androgen receptor with the antagonist flutamide (Fig. 4A). Prepubertal DHT exposure led to a dramatic decrease in type I fibers but this effect was not observed in DHT-exposed mice with skeletal muscle-specific deletion of the androgen receptor (Fig. 4B). These data strongly indicate that AR signaling is driving the decrease in type I muscle fibers in females.